# The SWELL1-LRRC8 complex regulates endothelial AKT-eNOS signaling and vascular function

Ahmad F Alghanem[1,2†], Javier Abello[3], Joshua M Maurer[1], Ashutosh Kumar[1], Chau My Ta[1], Susheel K Gunasekar[1], Urooj Fatima[4], Chen Kang[1], Litao Xie[1], Oluwaseun Adeola[4], Megan Riker[5], Macaulay Elliot-Hudson[4], Rachel A Minerath[4], Chad E Grueter[4], Robert F Mullins[5], Amber N Stratman[3], Rajan Sah[1,6]*

[1]Department of Internal Medicine, Cardiovascular Division, Washington University School of Medicine, St. Louis, United States; [2]Eastern Region, King Abdullah International Medical Research Center, King Saud bin Abdulaziz University for Health Sciences, Al Hasa, Saudi Arabia; [3]Department of Cell Biology and Physiology, Washington University in St. Louis, School of Medicine, St. Louis, United States; [4]Department of Internal Medicine, Cardiovascular Division, University of Iowa, Iowa City, United States; [5]Department of Ophthalmology, University of Iowa, Carver College of Medicine, Iowa City, United States; [6]Center for Cardiovascular Research, Washington University, St Louis, United States

*For correspondence:
rajan.sah@wustl.edu

Present address: [†]Eastern Region, King Abdullah International Medical Research Center, King Saud bin Abdulaziz University for Health Sciences, Al Hasa, Saudi Arabia

**Competing interests:** The authors declare that no competing interests exist.

**Abstract** The endothelium responds to numerous chemical and mechanical factors in regulating vascular tone, blood pressure, and blood flow. The endothelial volume-regulated anion channel (VRAC) has been proposed to be mechanosensitive and thereby sense fluid flow and hydrostatic pressure to regulate vascular function. Here, we show that the leucine-rich repeat-containing protein 8a, LRRC8A (SWELL1), is required for VRAC in human umbilical vein endothelial cells (HUVECs). Endothelial LRRC8A regulates AKT-endothelial nitric oxide synthase (eNOS)signaling under basal, stretch, and shear-flow stimulation, forms a GRB2-Cav1-eNOS signaling complex, and is required for endothelial cell alignment to laminar shear flow. Endothelium-restricted *Lrrc8a* KO mice develop hypertension in response to chronic angiotensin-II infusion and exhibit impaired retinal blood flow with both diffuse and focal blood vessel narrowing in the setting of type 2 diabetes (T2D). These data demonstrate that LRRC8A regulates AKT-eNOS in endothelium and is required for maintaining vascular function, particularly in the setting of T2D.

## Introduction

The endothelium integrates mechanical and chemical stimuli to regulate vascular tone, angiogenesis, blood flow, and blood pressure (*Cahill and Redmond, 2016*). Endothelial cells express a variety of mechanosensitive and mechanoresponsive ion channels that regulate vascular function (*Gerhold and Schwartz, 2016*), including TRPV4 (*Sonkusare et al., 2012*; *Earley et al., 2009*; *Zhang et al., 2009*; *Mendoza et al., 2010*) and Piezo1 (*Rode et al., 2017*; *Li et al., 2014*; *Coste et al., 2010*). The volume-regulated anion channel (VRAC) current is also prominent in endothelium and has been proposed to be mechanoresponsive (*Nilius and Droogmans, 2001*), to activate in response to fluid flow/hydrostatic pressure (*Barakat et al., 1999*) and to putatively regulate vascular reactivity. However, the molecular identity of this endothelial ion channel has remained a mystery for nearly two decades.

*Lrrc8a* (leucine-rich repeat-containing protein 8A, also known as SWELL1) encodes a transmembrane protein first described as the site of a balanced translocation in an immunodeficient child with

 

agammaglobulinemia and absent B-cells (*Sawada et al., 2003*; *Kubota et al., 2004*). Subsequent work revealed the mechanism for this condition to be due to impaired LRRC8A-dependent GRB2-PI3K-AKT signaling in lymphocytes, resulting in a developmental block in lymphocyte differentiation (*Kumar et al., 2014*). Thus, for ~11 years, LRRC8A was conceived of as a membrane protein that regulates PI3K-AKT mediated lymphocyte function (*Sawada et al., 2003*; *Kubota et al., 2004*). Although LRRC8A had been predicted to form a hetero-hexameric ion channel complex with other LRRC8 family members (*Abascal and Zardoya, 2012*), it was not until 2014 that LRRC8A was shown to form an essential component of the volume-regulated anion channel (VRAC) (*Qiu et al., 2014*; *Voss et al., 2014*), forming hetero-hexamers with LRRC8b-e (*Voss et al., 2014*; *Syeda et al., 2016*). Therefore, historically, LRRC8A was first described as a membrane protein that signaled via protein-protein interactions and then later found to form an ion channel signaling complex.

We showed previously that LRRC8A is an essential component of VRAC in adipocytes (*Zhang et al., 2017*) and skeletal myotubes (*Kumar et al., 2020*), which is required for insulin-PI3K-AKT signaling (*Zhang et al., 2017*; *Kumar et al., 2020*) to mediate adipocyte hypertrophy (*Zhang et al., 2017*), skeletal myotube differentiation (*Kumar et al., 2020*), skeletal muscle function (*Kumar et al., 2020*), and systemic glucose homeostasis (*Zhang et al., 2017*; *Kumar et al., 2020*; *Xie et al., 2017*; *Gunasekar et al., 2019*). The PI3K-AKT-endothelial nitric oxide synthase (eNOS) signaling pathway is central to transducing both mechanical stretch (*Hu et al., 2013*) and hormonal inputs (insulin) to regulate eNOS expression and activity, which, in turn, regulates vasodilation (blood flow and pressure), inhibits leukocyte aggregation, and limits proliferation of vascular smooth muscle cells (atherosclerosis). Indeed, insulin resistance is thought to be a systemic disorder in the setting of type 2 diabetes (T2D), affecting endothelium in addition to traditional metabolically important tissues, such as adipose, liver, and skeletal muscle (*Janus et al., 2016*; *Kearney et al., 2008*; *Muniyappa and Sowers, 2013*). In fact, insulin-resistant endothelium and the resultant impairment in PI3K-AKT-eNOS signaling have been proposed to underlie much of the endothelial dysfunction observed in the setting of obesity and T2D, predisposing to hypertension, atherosclerosis, and vascular disease (*Janus et al., 2016*; *Kearney et al., 2008*; *Muniyappa and Sowers, 2013*).

In this study, we demonstrate that VRAC is LRRC8A-dependent in endothelium, associates with GRB2, caveolin-1 (Cav1), eNOS, and regulates PI3K-AKT-eNOS signaling and flow-mediated endothelial cell orientation – suggesting that LRRC8 channel complexes link insulin and mechano-signaling in endothelium. LRRC8A-dependent AKT-eNOS, ERK1/2, and mTOR signaling influences blood pressure and vascular function in vivo, while impaired endothelial LRRC8A signaling predisposes to vascular dysfunction in the setting of diet-induced T2D.

## Results

### *Lrrc8a* functionally encodes VRAC current in endothelium

The volume-regulatory anion channel (VRAC) current has been measured and characterized in endothelial cells for decades but the molecular identity of this endothelial ion channel remains elusive (*Nilius and Droogmans, 2001*; *Barakat et al., 1999*; *Barakat, 1999*). To determine if the leucine-rich repeat-containing membrane protein LRRC8A recently identified in cell lines (*Qiu et al., 2014*; *Voss et al., 2014*) is required for VRAC in endothelial cells, as it is in adipocytes (*Zhang et al., 2017*), skeletal myoblasts (*Kumar et al., 2020*) pancreatic ß-cells (*Kang et al., 2018*; *Stuhlmann et al., 2018*), nodose neurons (*Wang et al., 2017*), and spermatozoa (*Lück et al., 2018*), we first confirmed robust LRRC8A protein expression by Western blot (*Figure 1A*) and immunostaining (*Figure 1B*) in human umbilical vein endothelial cells (HUVECs). LRRC8A protein expression is substantially reduced upon adenoviral transduction with a short-hairpin RNA directed to *Lrrc8a* (Ad-sh*Lrrc8a*-mCherry) as compared to a scrambled control (Ad-sh*Scr*-mCherry). Next, we measured hypotonically induced (210 mOsm) endothelial VRAC currents in HUVECs. These classic outwardly rectifying hypotonically induced VRAC currents are prominent in HUVECs, largely blocked by the VRAC current inhibitor 4-(2-butyl-6,7-dichloro-2-cyclopentyl-indan-1-on-5-yl) oxobutyric acid (DCPIB; *Figure 1C&D*), and significantly suppressed upon sh*Lrrc8a*-mediated LRRC8A knock-down (KD) (*Figure 1E&F*), consistent with LRRC8A functionally encoding endothelial VRAC.

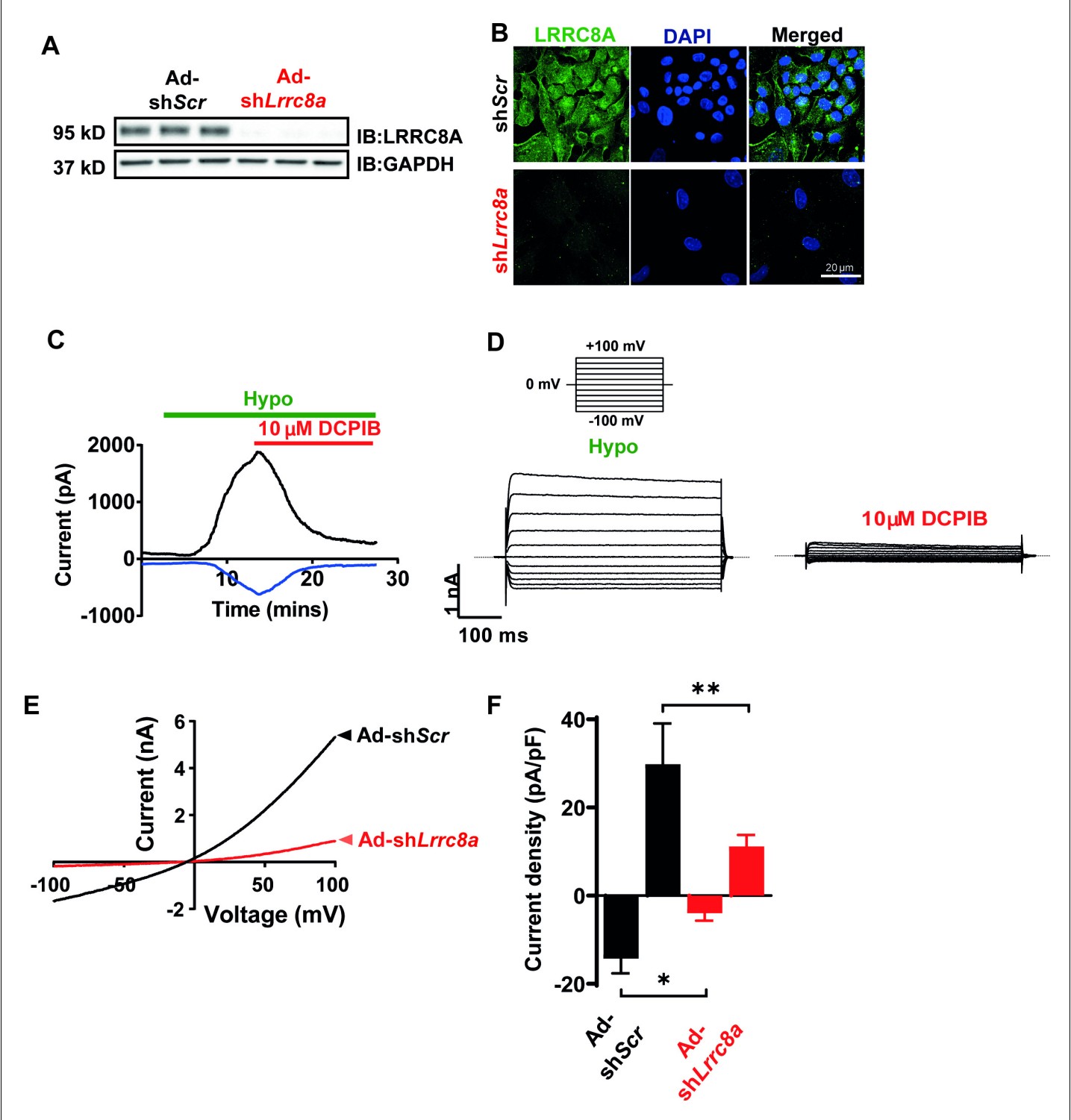

**Figure 1.** Leucine-rich repeat-containing protein 8a (LRRC8A) mediates volume-regulated anion channel (VRAC) currents in human umbilical vein endothelial cells (HUVECs). (**A**) LRRC8A Western blot in HUVECs transduced with adenovirus expressing a short-hairpin RNA directed to LRRC8A (Ad-sh*Lrrc8a*) compared to control scrambled short-hairpin RNA (Ad-sh*Scr*). GAPDH is used as loading control. (**B**) Immunofluorescence staining of the HUVECs transduced with Ad-sh*Lrrc8a* and Ad-sh*Scr*. (**C**) Current-time relationship of VRAC current (hypotonic, 210 mOsm) in Ad-sh*Scr* transduced HUVEC and co-application of 10 μM 4-(2-butyl-6,7-dichloro-2-cyclopentyl-indan-1-on-5-yl) oxobutyric acid (DCPIB). (**D**) Representative current traces upon hypotonic activation (left) during voltage steps (from -100 to +100 mV, shown in inset) and inhibition by DCPIB (right). (**E**) Current-voltage relationship of VRAC during voltage ramps from -100 mV to +100 mV after hypotonic swelling in HUVECs transduced with Ad-sh*Scr* and Ad-sh*Lrrc8a*.
*Figure 1 continued on next page*

*Figure 1 continued*

(F) Mean current outward and inward densities at +100 and -100 mV ($n_{shSCR}$=4 cells; $n_{shLrrc8a}$=6 cells). Data are shown as mean ± s.e.m. *p<0.05; **p<0.01; unpaired t-test for (F).

The online version of this article includes the following source data for figure 1:

**Source data 1.** Source data for *Figure 1F*.

## LRRC8A regulates PI3K-AKT-eNOS, ERK, and mTOR signaling in endothelium

Previous studies in adipocytes (*Zhang et al., 2017*) and skeletal muscle (*Kumar et al., 2020*) demonstrate that LRRC8A regulates insulin-PI3K-AKT signaling (*Zhang et al., 2017*; *Kumar et al., 2020*), adipocyte expansion (*Zhang et al., 2017*), skeletal muscle function (*Kumar et al., 2020*), and systemic glycemia, whereby LRRC8A loss-of-function induces an insulin-resistant pre-diabetic state (*Zhang et al., 2017*; *Kumar et al., 2020*; *Xie et al., 2017*). Insulin signaling is also important in regulating endothelium and vascular function (*Kearney et al., 2008*; *Muniyappa and Sowers, 2013*; *Duncan et al., 2008*). Moreover, insulin resistance in T2D is considered a systemic disorder and insulin-resistant endothelium is postulated to underlie impaired vascular function in T2D (*Kearney et al., 2008*; *Muniyappa and Sowers, 2013*). As LRRC8A is highly expressed in endothelium (*Figure 1*) and PI3K-AKT-eNOS signaling is critical for endothelium-dependent vascular function (*Morello et al., 2009*), we next examined AKT-eNOS, ERK1/2, and mTOR signaling in LRRC8A KD compared to control HUVECs under basal conditions. Basal phosphorylated AKT2 (pAKT2, *Figure 2A&D*), pAKT1 (*Figure 2A&E*), phosphorylated eNOS (p-eNOS) (*Figure 2A&C*), and pERK1/2 (*Figure 2A&F*) are abrogated in HUVECs upon LRRC8A KD, indicating that LRRC8A contributes to AKT-eNOS and ERK signaling in endothelium. Curiously, basal pS6 ribosomal protein, indicative of mTOR signaling, is augmented in LRRC8A KD HUVECs compared to control (*Figure 2A&G*), suggesting LRRC8A to be a negative regulator of mTOR in endothelium. As a complementary approach, we used siRNA mediated LRRC8A KD using a silencer select siRNA targeting *Lrrc8a* mRNA with a different sequence from sh*Lrrc8a*. siRNA mediated LRRC8A KD in HUVECs yielded nearly identical results to the shRNA KD approach (*Figure 2—figure supplement 1*). In summary, LRRC8A expression level regulates AKT, eNOS, ERK, and mTOR signaling in endothelium.

## LRRC8A interacts with GRB2, Cav1, and eNOS and mediates stretch-dependent eNOS signaling

In adipocytes, the mechanism of LRRC8A-mediated regulation of PI3K-AKT signaling involves LRRC8A/GRB2/Cav1 molecular interactions (*Zhang et al., 2017*). To determine if LRRC8A resides in a similar macromolecular signaling complex in endothelium, we immunoprecipitated (IP) endogenous GRB2 from HUVECs. Upon GRB2 IP, we detected LRRC8A protein in sh*Scr*-treated HUVECs and less LRRC8A upon GRB2 IP from sh*Lrrc8a*-treated HUVECs, consistent with an LRRC8A-GRB2 interaction (*Figure 3A&B*). In addition, with GRB2 IP, we also detected both Cav1 (*Figure 3A&B*) and eNOS (*Figure 3B*). These data suggest that endothelial LRRC8A resides in a signaling complex that includes GRB2, Cav1, and eNOS, consistent with the findings that GRB2 and Cav1 interact, and that Cav1 regulates eNOS via a direct interaction (*Ju et al., 1997*; *Venema et al., 1997*; *Goligorsky et al., 2002*). Also, GRB2 has been shown to regulate endothelial ERK, AKT, and JNK signaling (*Salameh et al., 2005*). Moreover, these data are also in line with the notion that caveoli form mechanosensitive microdomains (*Nassoy and Lamaze, 2012*; *Sinha et al., 2011*; *Sedding et al., 2005*) which regulate VRAC (*Trouet et al., 2001*; *Trouet et al., 1999*; *Egorov et al., 2019*), and VRAC can be activated by mechanical stimuli in a number of cell types, including endothelium (*Nilius and Droogmans, 2001*; *Barakat, 1999*; *Browe and Baumgarten, 2003*; *Browe and Baumgarten, 2006*; *Nakao et al., 1999*; *Romanenko et al., 2002*).

We also examined the relationship between LRRC8A and eNOS protein expression and localization in HUVECs by immunofluorescence (IF) staining (*Figure 3C*). Similar to observed by Western blot (*Figure 2A*, *Figure 2—figure supplement 1*), IF staining revealed that reductions in LRRC8A expression correlate with reduced eNOS expression (*Figure 3C–E*, *Figure 3—figure supplement 1*). Moreover, LRRC8A and eNOS co-localization in plasma membrane and perinuclear intracellular domains (*Figure 3C*, inset) were consistent with the IP data, revealing an LRRC8A-GRB2-Cav-eNOS

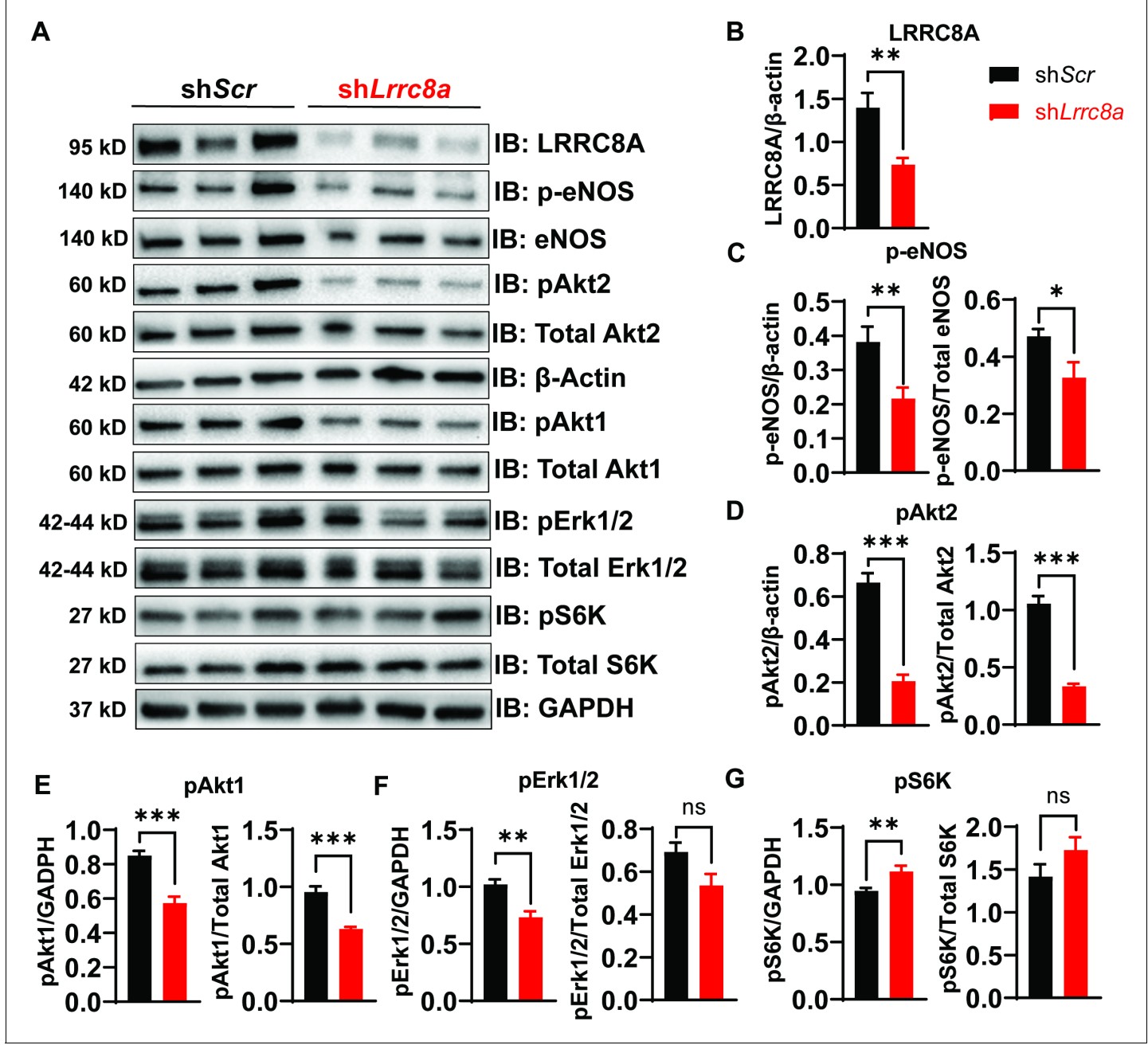

**Figure 2.** Leucine-rich repeat-containing protein 8a (LRRC8A) regulates PI3K-AKT-endothelial nitric oxide synthase (eNOS), ERK signaling in endothelium. (**A**) Western blots of LRRC8A, pAkt2, pAkt1, Akt2, Akt1, pErk1/2, Erk1/2, phosphorylated eNOS (p-eNOS), eNOS, pS6K ribosomal protein, S6K ribosomal protein, GAPDH, and ß-actin in Ad-sh*Scr* and Ad-sh*Lrrc8a* transduced human umbilical vein endothelial cells (HUVECs) under basal conditions. Quantification of LRRC8A/ß-actin (**B**), p-eNOS/ß-actin, p-eNOS/total eNOS (**C**), pAkt2/ß-actin, pAkt2/total Akt2 (**D**), pAkt1/GAPDH, pAkt1/total Akt1 (**E**), pERK1/2/GAPDH, pErk1/2/total Erk1/2 (**F**), pS6 ribosomal protein/GAPDH, and pS6K ribosomal protein/total S6K ribosomal protein (**G**). N = 6 independent experiments. Significance between the indicated groups in all blots was calculated using a two-tailed unpaired Student's t-test. Data are shown as mean ± s.e.m. *p<0.05; **p<0.01; ***p<0.001.

The online version of this article includes the following source data and figure supplement(s) for figure 2:

**Source data 1.** Source data for *Figure 2*.

**Figure supplement 1.** Leucine-rich repeat-containing protein 8a (LRRC8A) regulates PI3K-AKT-endothelial nitric oxide synthase (eNOS) and ERK signaling in endothelium.

**Figure supplement 1—source data 1.** Source data for *Figure 2—figure supplement 1*.

interaction (*Figure 3A&B*), and also with the previously described intracellular eNOS localization (*Fulton et al., 2002*).

Given that endothelial cells respond to stretch stimuli to regulate vascular tone via activation of eNOS (*Jufri et al., 2015*), we next examined the LRRC8A dependence of stretch-induced AKT, ERK1/2, and eNOS signaling in HUVECs (*Figure 4*), similar to several previous studies (*Hsu et al., 2010*; *Shi et al., 2007*; *Liu et al., 2003*; *Goettsch et al., 2009*). Stretch (5%) was sufficient to stimulate AKT1 and AKT2 signaling (*Figure 4A–C*), though not ERK1/2 signaling (*Figure 4D*) in HUVECs, and all were blunted in LRRC8A KD HUVECS (*Figure 4A–D*). Similarly, we observed abrogation of time-dependent p-eNOS signaling with 5% stretch in LRRC8A KD HUVECs compared to control (*Figure 4E&F*). Taken together, these data position LRRC8A as a regulator stretch-mediated PI3K-AKT-eNOS signaling in endothelium.

## Endothelial cell orientation to the direction of shear flow is impaired upon LRRC8A KD

To evaluate the requirement of LRRC8A for endothelial cell responses to physiological mechanical stimuli relevant to endothelium, we examined the LRRC8A dependence of HUVEC orientation to laminar shear flow. In response to laminar shear flow at 15 dynes/cm$^2$ for 24 hr (*Figure 5A*), siControl transfected HUVECs oriented well to the direction of flow (*Figure 5B–F* and *Figure 5—videos 1* and *2*), while LRRC8A KD HUVEC exhibited impaired flow-mediated alignment (*Figure 5B–F* and *Figure 5—videos 3* and *4*). These LRRC8A-dependent differences were also apparent in the degree of impairment of HUVEC elongation to the direction of fluid flow (*Figure 5G*) and in the velocity of movement against the direction of flow (*Figure 5H&I*).

Next, we asked whether shear-flow-mediated eNOS phosphorylation is LRRC8A-dependent, as observed with stretch-mediated signaling (*Figure 4*). We stimulated si*Scr* and si*Lrrc8a* transfected HUVECs with either no flow, or laminar shear flow for 2 or 24 hr and then immunostained for p-eNOS (*Figure 5J*). We observed marked p-eNOS induction in response to shear flow after both 2 and 24 hr in siControl transfected HUVECs, and this was significantly abrogated upon LRRC8A KD (*Figure 5J*). Interestingly, this shear-flow stimulated p-eNOS displayed significant nuclear localization, consistent with prior reports (*Feng et al., 1999*; *Andersson and Brittebo, 2012*; *Wang et al., 2019*; *Gobeil et al., 2006*). The same nuclear distribution was also observed using a different p-eNOS antibody (*Figure 5—figure supplement 1*). We quantified the intensity of p-eNOS staining in two ways: p-eNOS excluding the nuclear signal (*Figure 5K,L*) and p-eNOS in the entire cell (*Figure 5M,O*). In both cases, we observed significant shear-flow-mediated p-eNOS induction in si*Scr* HUVECs, after both 2 hr and 24 hr flow, and this was reduced in LRRC8A KD HUVEC (*Figure 5K–O*), consistent with LRRC8A-dependent flow-mediated p-eNOS signaling.

## Genome-wide mRNA profiling reveals LRRC8A-dependent regulation of multiple processes involved in angiogenesis, atherosclerosis, and vascular function

Genome-wide transcriptome analysis of LRRC8A KD HUVEC compared to control (RNA sequencing) revealed multiple pathways enriched regulating angiogenesis, migration, and tumorigenesis, including GADD45, IL-8, p70S6K (mTOR), TREM1, angiopoietin, and HGF signaling (*Figure 6*, *Supplementary file 1* and *2*). Pathways linked to cell adhesion and renin-angiotensin signaling are also enriched – both pathways and processes that are known to be altered in vasculature in the setting of atherosclerosis and T2D. Also, notable are statistically significant increases in VEGFA (1.6-fold) and CD31 (2.0-fold) expression in LRRC8A KD HUVECs (*Figure 6C*), both of which are pro-angiogenic and associated with mTORC1 hyperactivation (*Ding et al., 2018*). CD36 is increased 3.4-fold and is also pro-angiogenic (*Silverstein and Febbraio, 2009*) and pro-atherosclerotic (*Park, 2014*; *Harb et al., 2009*; *Kennedy et al., 2009*). Moreover, the trends toward reduced eNOS protein expression observed upon LRRC8A KD in HUVECs (*Figure 2A*, *Figure 2—figure supplement 1*, *Figure 3C&E*, *Figure 3—figure supplement 1*) are also associated with twofold reduced eNOS mRNA expression (NOS3, *Figure 6C*). Notably, *Lrrc8a* mRNA expression in HUVECs is higher than many classical endothelial markers expressed in endothelium (EPOR, CDH5, CD36, VWF, PECAM1, KDR, VEGFA, *Figure 6C*). HUVEC *Lrrc8a* mRNA expression is also 2.5-fold higher than Piezo1 (*Figure 6C*), a mechanosensitive ion channel (*Coste et al., 2010*; *Coste et al., 2012*) with a

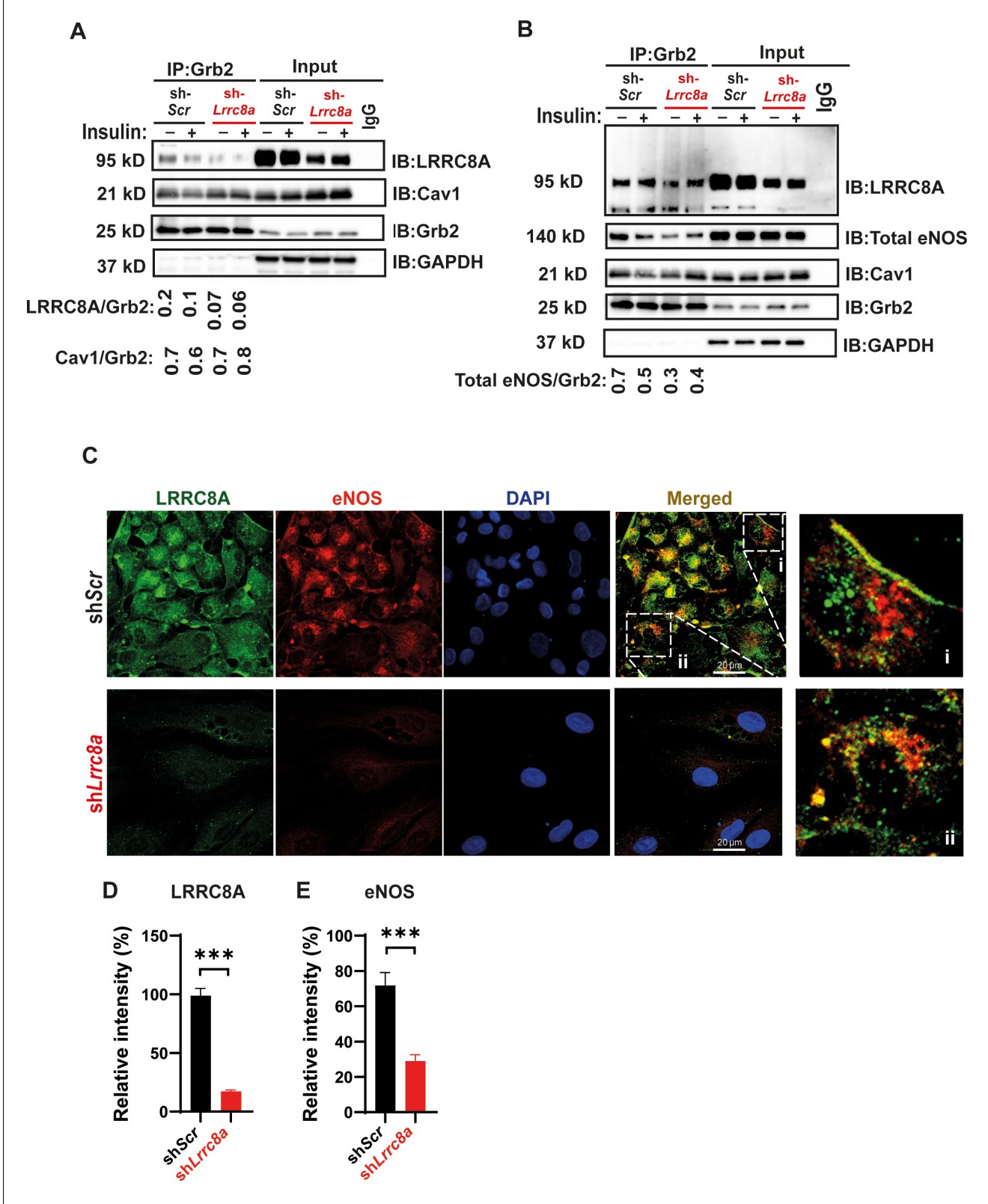

**Figure 3.** Leucine-rich repeat-containing protein 8A (LRRC8A) interacts with Grb2, Cav1, and endothelial nitric oxide synthase (eNOS) in human endothelium. (**A**) GRB2 immunoprecipitation from Ad-sh*Scr* and Ad-sh*Lrrc8a* transduced human umbilical vein endothelial cells (HUVECs) and immunoblot with LRRC8A, Cav1, and GRB2 antibodies. Densitometry values for GRB2 co-immunoprecipitated LRRC8A (LRRC8A/GRB2) and GRB2 co-immunoprecipitated Cav1 (Cav1/GRB2). GAPDH serves as loading control for input samples. (**B**) GRB2 immunoprecipitation from Ad-sh*Scr* and Ad-

*Figure 3 continued on next page*

Figure 3 continued

sh*Lrrc8a* transduced HUVECs and immunoblot with LRRC8A, eNOS, Cav1, and GRB2 antibodies. Insulin stimulation: 100 nM insulin for 10 min. Densitometry values for GRB2 co-immunoprecipitated eNOS (eNOS/GRB2). Representative blots from three independent experiments. (C) Representative endogenous LRRC8A and eNOS immunofluorescence staining in Ad-sh*Scr* and Ad-sh*Lrrc8a* transduced HUVECs. Representative image from six independent experiments. (D–E) Quantification of LRRC8A (D, n = 6) and eNOS (E, n = 6) immunofluorescence staining upon LRRC8A knock-down (KD). Evidence of LRRC8A-eNOS co-localization (C, insets) in plasma membrane (i) and perinuclear regions (ii). Significance between the indicated groups is calculated using an unpaired two-tailed Student's t-test. p-Values are indicated on figures. Data are shown as mean ± s.e.m. ***p<0.001. The online version of this article includes the following source data and figure supplement(s) for figure 3:

**Source data 1.** Source data for *Figure 3D*.
**Source data 2.** Source data for *Figure 3E*.
**Figure supplement 1.** Leucine-rich repeat-containing protein 8a (LRRC8A) co-localizes with endothelial nitric oxide synthase (eNOS) and regulates eNOS expression.

well-established role in endothelial biology and vascular function (*Rode et al., 2017*; *Li et al., 2014*), and Piezo1 mRNA expression does not change upon LRRC8A KD. Also, *Lrrc8a* mRNA expression is the highest among all *LRRC8* genes expressed in HUVECs, 3-fold higher than *Lrrc8a* mRNA in 3T3-F442A adipocytes (*Zhang et al., 2017*), 15-fold higher than C2C12 myotubes (*Kumar et al., 2020*; *Figure 6D*), consistent with the prominent endogenous $I_{Cl,SWELL}$ currents present in HUVEC (*Figure 1C–F*).

## Endothelial-targeted LRRC8A KO mice exhibit mild angiotensin-II stimulated hypertension and impaired retinal blood flow in the setting of T2D

To examine the functional consequences of endothelial LRRC8A ablation in vivo, we generated endothelial-targeted *Lrrc8a* knock-out mice (e*Lrrc8a* KO) by crossing *Lrrc8a* floxed mice (*Zhang et al., 2017*; *Kumar et al., 2020*; *Kang et al., 2018*) with the endothelium-restricted CDH5-Cre mouse (CDH5-Cre; *Lrrc8a* $^{fl/fl}$; *Figure 7A*). Patch-clamp recordings of primary endothelial cells isolated from WT and e*Lrrc8a* KO mice (*Figure 7B*) reveal robust hypotonically activated currents (Hypo, 210 mOsm) in WT endothelial cells, which are DCPIB inhibited (*Figure 7C&E*), while e*Lrrc8a* KO endothelial cells exhibit markedly reduced hypotonically activated currents (*Figure 7D&E*). Consistent with this, IF staining of aortic ring explants from WT and e*Lrrc8a* KO mice confirmed LRRC8A ablation from CD31+ primary endothelial cells (*Figure 7F&G*).

Based on our findings that LRRC8A regulates AKT-eNOS signaling in HUVECs in vitro, we next examined p-eNOS in aortic endothelium by IF staining in WT and LRRC8A KO mice. Similar to results in LRRC8A KD HUVEC, we found reduced p-eNOS intensity in aortic endothelium of e*Lrrc8a* KO mice as compared to WT mice (*Figure 7H,I*).

Since eNOS signaling is central to blood pressure regulation, we next examined blood pressures in e*Lrrc8a* KO mice compared to WT controls (*Lrrc8a* $^{fl/fl}$ mice). Male mice exhibited no significant differences in systolic blood pressure under basal conditions (*Figure 7J*), while female mice were mildly hypertensive relative to WT mice (*Figure 7K*). However, after 4 weeks of angiotensin-II (Ang II) infusion, male e*Lrrc8a* KO mice developed exacerbated systolic hypertension as compared to Ang II-treated WT mice (*Figure 7L*). These data are consistent with endothelial dysfunction and impaired vascular relaxation in e*Lrrc8a* KO mice, resulting in a propensity for systolic hypertension.

As endothelial dysfunction may also result in impaired blood flow, we performed retinal imaging during intraperitoneal (i.p.) injection of fluorescein to assess retinal vessel blood flow and morphology in WT and e*Lrrc8a* KO mice. Mice raised on a regular diet had mild, non-significant impairments in retinal blood flow, based on the relative rate of rise of the fluorescein signal in retinal vessels (*Figure 8—figure supplement 1*). There was evidence of mild focal narrowing of retinal vessels in e*Lrrc8a* KO as compared to WT mice (*Figure 8—figure supplement 1A,D&E*), with no significant differences in other parameters (*Figure 8—figure supplement 1F–K*). In mice raised on high-fat high-sucrose (HFHS) diet, retinal blood flow was more severely impaired (*Figure 8A–C*) with significant focal and diffuse retinal vessel narrowing in e*Lrrc8a* KO mice compared to WT mice (*Figure 8A, F–H*, *Figure 8—figure supplement 2*, *Figure 8—video 1*), and this relative difference was markedly worse in female compared to male mice. These findings are all consistent with endothelial dysfunction and impaired retinal vessel vasorelaxation due to reduced eNOS expression and activity,

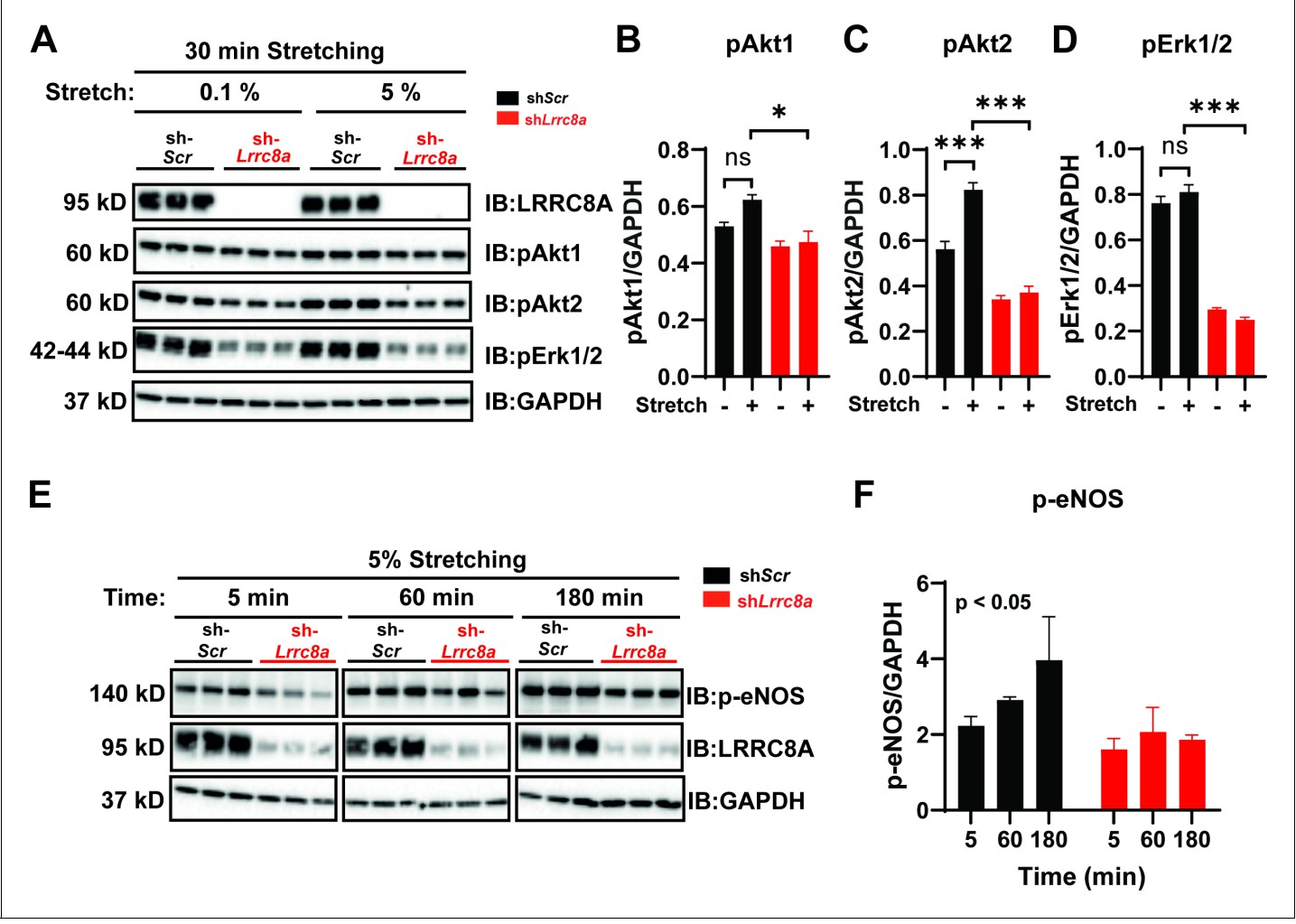

**Figure 4.** Leucine-rich repeat-containing protein 8a (LRRC8A) is required for intact stretch-induced AKT-endothelial nitric oxide synthase (eNOS) signaling. (**A**) Western blot of LRRC8A, pAKT1, pAKT2, pERK1/2 in response to 30 min of 0.1% and 5% static stretch in Ad-sh*Scr* (n = 3) and Ad-sh*Lrrc8a* (n = 3) transduced human umbilical vein endothelial cells (HUVECs). GAPDH is used as a loading control. (**B–D**) Densitometry quantification from A of pAKT1 (**B**), pAKT2 (**C**), and pErk1/2 (**D**). (**E**) Western blot of peNOS, LRRC8A, in response to 5% static stretching for 5, 60, and 180 min in Ad-sh*Scr* (n = 3) and Ad-sh*Lrrc8a* (n = 3) transduced HUVECs. (**F**) Densitometry quantification from E of eNOS. GAPDH is used as a loading control. Statistical significance between the indicated group is calculated using one-way analysis of variance (ANOVA), Tukey's multiple comparisons test for (**B–D**), and two-way ANOVA for (**F**) with p-value in upper left corner. Data are shown as mean ± s.e.m. *p<0.05; **p<0.01; ***p<0.001.
The online version of this article includes the following source data for figure 4:

Source data 1. Source data for *Figure 4F*.
Source data 2. Source data for *Figure 4B,C, D*.

particularly in the setting of HFHS diet. Also consistent with impaired eNOS activity are reductions in vessel number (*Figure 8E*), vessel surface area (*Figure 8H*), number of end points (*Figure 8K*), branching index (*Figure 8L*), and increased lacunarity (*Figure 8J*). These parameters are all sugges-tive of diabetes-induced retinal vessel dysfunction in the e*Lrrc8a* KO mice, consistent with the loss of eNOS activity that is expected when insulin signaling is compromised (*Brooks et al., 2001*; *Kondo et al., 2003*). Notably, both WT and e*Lrrc8a* KO mice were found to be equally glucose-intol-erant and insulin-resistant (*Figure 8—figure supplement 3*), indicating that these differences in microvascular dysfunction were not due to increased hyperglycemia and more severe diabetes in e*Lrrc8a* KO mice. However, while there were no differences in mean body weight or fat mass between WT and e*Lrrc8a* KO males raised on HFHS diet (*Figure 8—figure supplement 4*), female e*Lrrc8a* KO mice had higher body weights than WT female mice, and this was driven by an increase

in total fat mass (*Figure 8—figure supplement 4*). Taken together, our findings reveal that LRRC8A is highly expressed in endothelium and functionally encodes endothelial VRAC. LRRC8A regulates basal, stretch, and shear-flow-mediated ERK, AKT-eNOS signaling, forms an LRRC8A-GRB2-Cav1-eNOS signaling complex, and regulates vascular function in vivo.

## Discussion

Our findings demonstrate that the LRRC8 hetero-hexamer functionally encodes endothelial VRAC, whereby LRRC8A associates with GRB2 and Cav1 and positively regulates PI3K-AKT-eNOS and ERK1/2 signaling. LRRC8A depletion in HUVECs reduces pAKT2, pAKT1, p-eNOS, and pERK1/2 under basal conditions, and abrogates both stretch and shear-flow stimulated p-eNOS stimulation. We also discovered the ability of HUVECs to align along the direction of laminar shear flow is markedly impaired in LRRC8A KD cells – suggesting these cells have an impaired ability to either sense or respond to laminar shear flow (or both). These data reveal that LRRC8A-mediated PI3K-AKT signaling is conserved in endothelium, similar to previous observations in adipocytes (*Zhang et al., 2017*) and skeletal myotubes (*Kumar et al., 2020*), and in turn positively regulates eNOS expression and activity. Consistent with this mechanism, endothelial-targeted *Lrrc8a* ablation in vivo predisposes to microvascular dysfunction in the setting of T2D and to hypertension in response to Ang II infusion. These results are in line with the notion that LRRC8A depleted endothelium contributes to an insulin-resistant state in which impaired PI3K-AKT-eNOS signaling results in a propensity for vascular dysfunction (*Kearney et al., 2008*; *Muniyappa and Sowers, 2013*). Insulin-mediated regulation of NO is physiologically (*Zeng and Quon, 1996*; *Zeng et al., 2000*; *Montagnani et al., 2002*) and pathophysiologically (*Steinberg et al., 1996*) important, as NO has vasodilatory (*Quillon et al., 2015*; *Palmer et al., 1987*), anti-inflammatory (*Kataoka et al., 2002*), antioxidant (*Clapp et al., 2004*), and antiplatelet effects (*Schäfer et al., 2004a*; *Schäfer et al., 2004b*; *Radomski et al., 1987a*; *Radomski et al., 1987b*). Impaired NO-mediated vascular reactivity is a predictor of future adverse cardiac events (*Schächinger et al., 2000*) and portends increased risk of atherosclerosis (*Bugiardini et al., 2004*). Consistent with these NO-mediated effects, the RNA sequencing data derived from LRRC8A KD HUVECs revealed enrichment in inflammatory, cell adhesion, and proliferation pathways (GADD45, IL-8, mTOR, TREM1 signaling) that may arise from LRRC8A-mediated dysregulation of eNOS activity. Moreover, *Lrrc8a* mRNA is highly expressed in HUVECs relative to adipocyte and myotube cell lines examined previously (*Zhang et al., 2017*; *Kumar et al., 2020*), higher than many endothelial genes, and higher than other mechanosensitive (Piezo1) and mechanoresponsive (TRPV4) ion channels with well-established roles in endothelial biology (*Sonkusare et al., 2012*; *Rode et al., 2017*; *Li et al., 2014*; *Cappelli et al., 2019*; *Dalsgaard et al., 2016*; *Dunn et al., 2013*; *Earley et al., 2005*; *Harraz et al., 2018*; *Mercado et al., 2014*; *Sonkusare et al., 2014*).

In addition to reductions in AKT-eNOS signaling, LRRC8A depletion in HUVECs also reduced ERK1/2 signaling. This decrease in pERK1/2 suggests impaired MAPK signaling which is connected to the insulin receptor by GRB2-SOS. Indeed, we also found that LRRC8A and GRB2 interact in HUVECs, and this may provide the molecular mechanism for the observed defect in ERK signaling. Interestingly, GRB2-MAPK signaling is thought to promote angiogenesis, migration, and proliferation (*Zhao et al., 2011*), so reductions in ERK1/2 signaling would be predicted to inhibit these processes. One hypothesized mechanism of LRRC8A-mediated regulation of ERK1/2 and AKT2-eNOS signaling involves non-conductive protein-protein signaling via the LRRD as described previously in adipocytes (*Zhang et al., 2017*; *Gunasekar et al., 2019*). An alternative hypothesis is that LRRC8A-mediated VRAC channel activity may alter endothelial membrane potential to regulate other endothelial ion channels, such as Piezo1 or TRPV4, and downstream calcium signaling to influence downstream signaling. Future studies will further delineate the molecular mechanisms of LRRC8A regulation of intracellular signaling in endothelial cells and vascular function, and also directly explore whether endothelial LRRC8A-mediated VRAC channel complexes are themselves mechanoresponsive, as suggested in prior studies in other cell types (*Browe and Baumgarten, 2003*; *Browe and Baumgarten, 2006*).

Another curious observation that warrants further exploration is the reduction in basal *Akt2* and *NOS2* (eNOS) mRNA expression upon LRRC8A KD in HUVECs, which is also observed at the protein level. We can only speculate that LRRC8A is somehow regulating *Akt2* and *eNOS* gene expression.

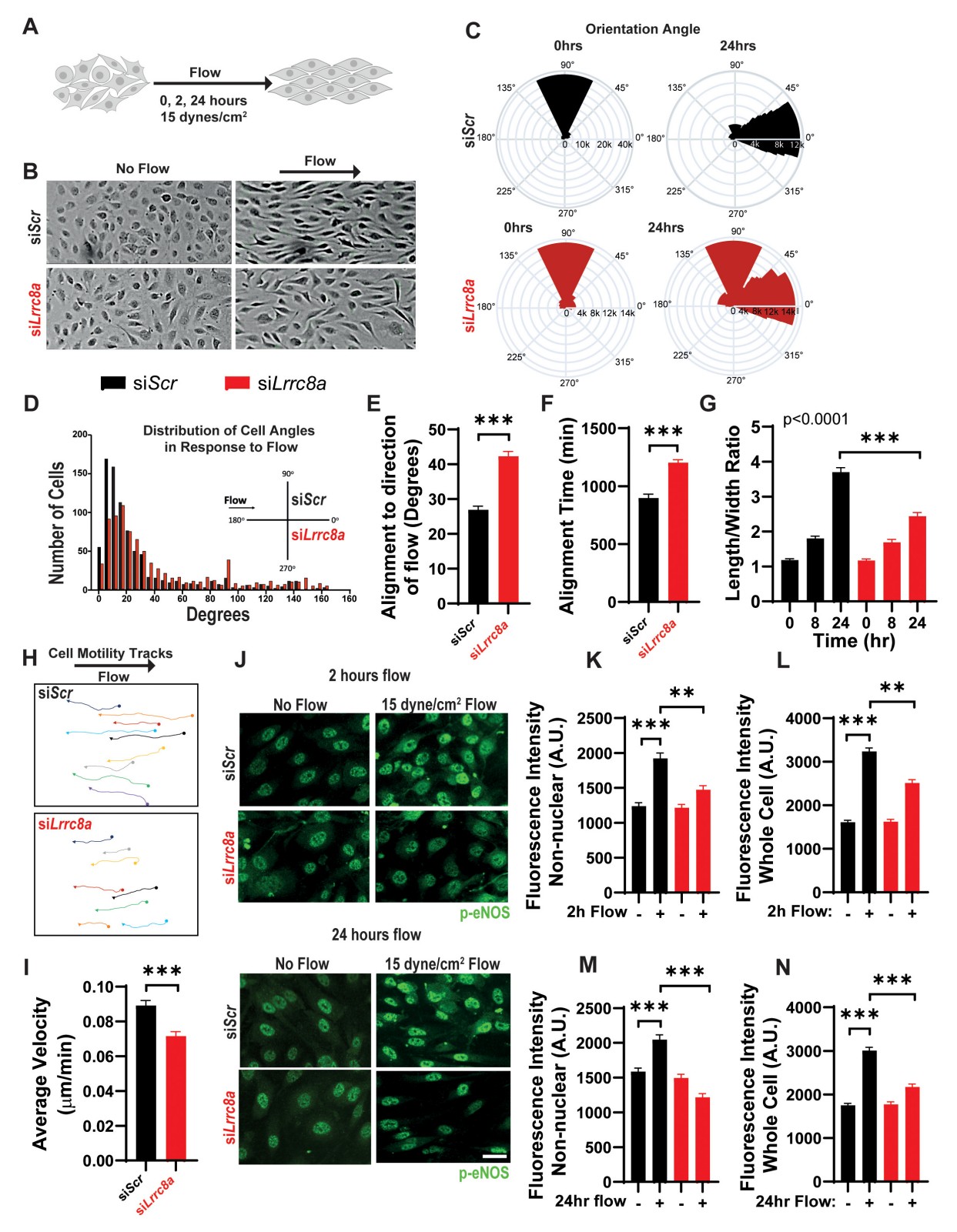

**Figure 5.** Leucine-rich repeat-containing protein 8a (LRRC8A) depletion impairs human umbilical vein endothelial cell (HUVEC) alignment to the direction of shear flow. (**A**) Cartoon of depicting the laminar shear flow applied to HUVEC at 15 dyne/cm² for 0, 2, and 24 hr. (**B**) Bright field image of HUVECs transfected with si*Scr* (top) and si*Lrrc8a* (bottom) with no flow (left) and after 24 hr of flow (right). (**C**) Distributions of angle of HUVEC orientation relative to the direction of flow at 0 hr and after 24 hr of laminar shear flow in si*Scr*-treated cells (top, 0 hr: n = 800; 24 hr: n = 800) and

*Figure 5 continued on next page*

*Figure 5 continued*

si*Lrrc8a* treated cells (bottom, 0 hr: n = 600; 24 hr: 800). Cuboidal or symmetrical cells at 0 hr were measured as 90° to the direction of flow. (D) Distributions of angle of HUVEC orientation relative to the direction of flow at 24 hr si*Scr* (black, n = 803) and si*Lrrc8a* treated cells (red, n = 803). (E) Mean alignment to direction of flow of si*Scr* (black, n = 803) and si*Lrrc8a* treated cells (red, n = 803). (F) Mean time required for HUVEC to align to within 20° of the direction of shear flow in si*Scr* (black, n = 110) and si*Lrrc8a* treated cells (red, n = 110). (G) Mean HUVEC length-width ratio over time upon stimulation with shear flow in si*Scr* (black, n = 157) and si*Lrrc8a* treated cells (red, n = 157). (H) Representative HUVEC real-time cell tracking during the 24 hr period of shear flow. Arrow heads represent the direction of travel against the direction of shear flow. (I) Average velocity calculated from the mean distance traveled over the 24 hr period of shear flow ($n_{siScr}$ = 77; $n_{siLrrc8a}$ = 78). (J) Representative phosphorylated endothelial nitric oxide synthase (p-eNOS) immunofluoresence (green) under conditions of no flow (n = 100), 2 hr (n = 100, top) and no flow (n = 110), 24 hr flow (n = 110, bottom). (K–L) Quantification of non-nuclear p-eNOS immunofluoresence (K) and whole-cell p-eNOS (L) under conditions of no flow (n = 100) and 2 hr flow (n = 100). (M–N) Quantification of non-nuclear p-eNOS immunofluoresence (M) and whole-cell p-eNOS (N) under conditions of no flow (n = 110) and 24 hr flow (n = 110). Statistical significance between the indicated values is calculated using a two-tailed Student's t-test for (E), (F), and (I), using one-way analysis of variance (ANOVA), Tukey's multiple comparisons test for (K), (L), (M), and (N), and using two-way ANOVA with Bonferroni multiple comparison test in (G). In (G), p<0.001 for the time dependence of si*Scr* versus si*Lrrc8a* treated cells, and p<0.001 for the 24 hr timepoint. Error bars represent mean ± s.e.m. n = 3, independent experiments. **p<0.01 and ***p<0.001.

The online version of this article includes the following video, source data, and figure supplement(s) for figure 5:

**Source data 1.** Source data for *Figure 5E*.
**Source data 2.** Source data for *Figure 5F*.
**Source data 3.** Source data for *Figure 5G*.
**Source data 4.** Source data for *Figure 5I*.
**Source data 5.** Source data for *Figure 5K*.
**Source data 6.** Source data for *Figure 5L*.
**Source data 7.** Source data for *Figure 5M*.
**Source data 8.** Source data for *Figure 5N*.
**Figure supplement 1.** Leucine-rich repeat-containing protein 8a (LRRC8A) knock-down abrogates shear-flow stimulated phosphorylated endothelial nitric oxide synthase (p-eNOS) in human umbilical vein endothelial cells (HUVECs).
**Figure supplement 1—source data 1.** Source data for *Figure 5—figure supplement 1B*.
**Figure supplement 1—source data 2.** Source data for *Figure 5—figure supplement 1C*.
**Figure 5—video 1.** Bright field time-lapse video of si*Scr* transfected human umbilical vein endothelial cell (HUVEC) responding to shear flow (direction of flow from left to right) over a 24 hr period.
https://elifesciences.org/articles/61313#fig5video1
**Figure 5—video 2.** Bright field time-lapse video of si*Scr* transfected human umbilical vein endothelial cell (HUVEC) responding to shear flow (direction of flow from left to right) over a 24 hr period.
https://elifesciences.org/articles/61313#fig5video2
**Figure 5—video 3.** Bright field time-lapse video of si*Lrrc8a* transfected human umbilical vein endothelial cell (HUVEC) responding to shear flow (direction of flow from left to right) over a 24 hr period.
https://elifesciences.org/articles/61313#fig5video3
**Figure 5—video 4.** Bright field time-lapse video of si*Lrrc8a* transfected human umbilical vein endothelial cell (HUVEC) responding to shear flow (direction of flow from left to right) over a 24 hr period.
https://elifesciences.org/articles/61313#fig5video4

Both PI3K and ERK1/2 signaling pathways have been described to regulate total eNOS expression (*Wu, 2002*), and we observe reductions in both PI3K and ERK1/2 signaling in LRRC8A KD HUVECs, so reductions in eNOS expression are certainly consistent with disruptions in these signaling pathways. Consistent with this conserved biology with respect to LRRC8A signaling, we also observed significant twofold reductions in AKT2 expression upon *Lrrc8a* deletion in both 3T3-F442A adipocytes (*Zhang et al., 2017*) and C2C12 myotubes (*Kumar et al., 2020*), in addition to reductions in NOS2 (iNOS) (*Kumar et al., 2020*) in adipocytes and NOS1 (neuronal NOS) in C2C12 myotubes (*Kumar et al., 2020*). Collectively, these findings across multiple cell types support a PI3K and ERK1/2 mediated mechanism of regulation of eNOS and AKT2 gene expression.

The phenotype of endothelial-targeted *Lrrc8a* KO (e*Lrrc8a* KO) mice are consistent with a reduction in eNOS and p-eNOS as these mice exhibit mild hypertension at baseline (females) and exacerbated hypertension in response to chronic angiotensin infusion, suggesting a modulatory effect on vascular reactivity. Similarly, retinal blood flow is only mildly impaired in e*Lrrc8a* KO mice raised on a regular diet, with some evidence of microvascular disease. However, both retinal blood flow and retinal vessel morphology are markedly impaired in obese-T2D, insulin-resistant e*Lrrc8a* KO mice raised on an HFHS diet compared to controls. This is consistent with a synergistic role of endothelial

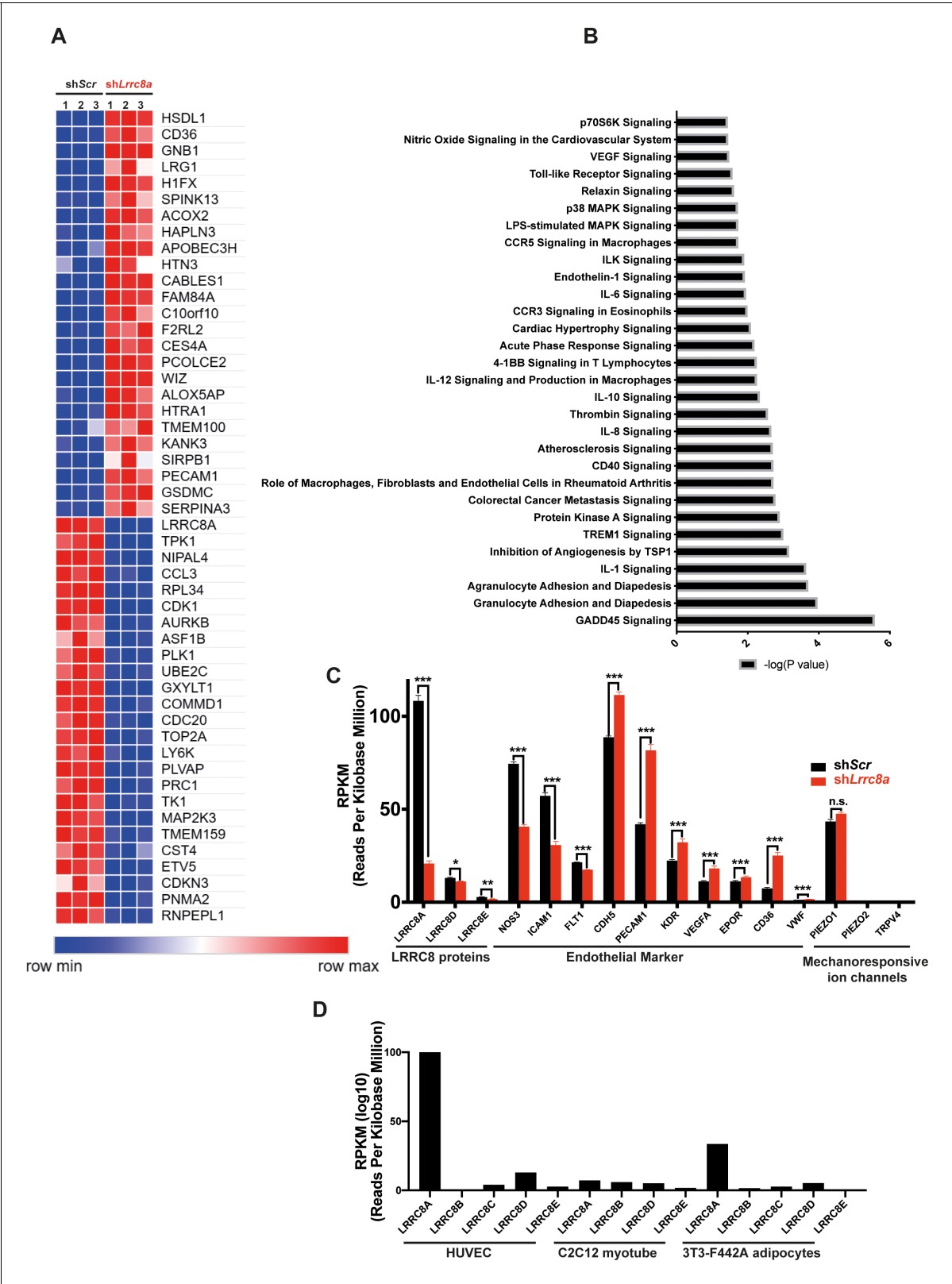

**Figure 6.** RNA sequencing of Ad-sh*Scr* and Ad-sh*Lrrc8a* transduced human umbilical vein endothelial cells (HUVECs). (**A**) Heatmap analysis displaying top 25 upregulated or 25 downregulated genes between sh*Scr* and sh*Lrrc8a*. (**B**) IPA canonical pathway analysis of genes significantly regulated by sh*Lrrc8a* in comparison to sh*Scr* n = 3 for each group. For analysis with Ingenuity Pathway Analysis (IPA), Fragments Per Kilobase of transcripts per Million mapped reads (FPKM) cutoffs of 1.5, fold change of 1.5, and false discovery rate <0.05 were utilized for significantly differentially regulated

*Figure 6 continued on next page*

*Figure 6 continued*

genes. (C) Expression levels of select *LRRC8* mRNA, classic endothelial markers, and mechanoresponsive ion channels in sh*Scr* and sh*Lrrc8a*-treated HUVECs. (D) Expression levels *LRRC8a-e* mRNA in HUVECs compared to data previously published from C2C12 myotubes (*Kumar et al., 2020*) and differentiated 3T3-F442A adipocytes (*Zhang et al., 2017*). For (C), statistical significance between the indicated values was calculated using an unpaired two-tailed Student's t-test. Error bars represent mean ± s.e.m. n = 3, independent experiments. *p<0.05; **p<0.01; ***p<0.001.

The online version of this article includes the following source data for figure 6:

**Source data 1.** Source data for *Figure 6C*.
**Source data 2.** Source data for *Figure 6D*.

e*Lrrc8a* ablation and T2D/obesity in the pathogenesis of vascular disease. Our results suggest that reductions in LRRC8A signaling may contribute to impaired vascular function observed in humans in response to insulin and/or shear stress in the setting of obesity (*Arcaro et al., 1999*; *Tack et al., 1998*; *Westerbacka et al., 1999*; *Williams et al., 2006*) and insulin resistance (*Murphy et al., 2007*).

# Materials and methods

## Key resources table

| Reagent type (species) or resource | Designation | Source or reference | Identifiers | Additional information |
|---|---|---|---|---|
| Genetic reagent (*Mus musculus*) | eLrrc8a KO (*Lrrc8a*$^{fl/fl}$) | This paper | Sah lab | SWELL1 is a regulator of adipocyte size, insulin signaling, and glucose homeostasis (*Zhang et al., 2017*) |
| Strain, strain background (*Mus musculus*) | *CDH5-Cre* | Gift from Dr. Kaikobad Irani | | Vikram, A, *Nature Communications* 2016 |
| Strain, strain background (*Mus musculus*) | Rosa26-tdTomato | Jackson lab | Jax 007914 RRID:IMSR_JAX:007914 | |
| Other | HUVECs | ATCC | Cat#CRL-2922 RRID:CVCL_3901 | Human primary cells |
| Biological sample (*Mus musculus*) | Endothelial primary cell | *Lrrc8a*$^{fl/fl}$ | | Freshly isolated from *Lrrc8a*$^{fl/fl}$ mice |
| Antibody | Anti-ß-actin (rabbit monoclonal) | Cell Signaling | Cat#8457s, RRID:AB_10950489 | WB (1:2000) |
| Antibody | Anti-total Akt (rabbit monoclonal) | Cell Signaling | Cat#4685s, RRID:AB_2225340 | WB (1:1000) |
| Antibody | Anti-Akt1 (rabbit monoclonal) | Cell Signaling | Cat#2938s, RRID:AB_915788 | WB (1:1000) |
| Antibody | Anti-p-eNOS (rabbit monoclonal) | Cell Signaling | Cat#9571, RRID:AB_329837 | WB (1:1000) IF (1:200) |
| Antibody | Anti-p-eNOS (rabbit polyclonal) | Invitrogen | Cat#PA5-104858, RRID:AB_2816331 | IF (1:200) |
| Antibody | Anti-Akt2 (rabbit monoclonal) | Cell Signaling | Cat#3063s, RRID:AB_2225186 | WB (1:1000) |
| Antibody | Anti-p-AS160 (rabbit polyclonal) | Cell Signaling | Cat#4288s, RRID:AB_10545274 | WB (1:1000) |
| Antibody | Anti-total eNOS (rabbit polyclonal) | Cell Signaling | Cat#32027, RRID:AB_2728756 | WB (1:1000) |
| Antibody | Anti-PECAM (rabbit monoclonal) | Sigma | Cat#SAB4502167 RRID:AB_10762600 | WB (1:1000) |
| Antibody | Anti-VEGFR2 (goat polyclonal) | R and D Systems | Cat#BAF357 RRID:AB_356414 | WB (1:1000) |

*Continued on next page*

*Continued*

| Reagent type (species) or resource | Designation | Source or reference | Identifiers | Additional information |
|---|---|---|---|---|
| Antibody | Anti- pS6 ribosomal (rabbit monoclonal) | Cell Signaling | Cat#5364s, RRID:AB_10694233 | WB (1:2000) |
| Antibody | Anti- GAPDH (rabbit monoclonal) | Cell Signaling | Cat#5174s, RRID:AB_1062202 | WB (1:2000) |
| Antibody | Anti-pErk1/2 (rabbit polyclonal) | Cell Signaling | Cat#9101s, RRID:AB_331772 | WB (1:1000) |
| Antibody | Anti-Total Erk1/2 (rabbit polyclonal) | Cell Signaling | Cat#9102s, RRID:AB_330744 | WB (1:1000) |
| Antibody | Anti-Grb2 (mouse monoclonal) | BD | Cat#610111s, RRID:AB_397517 | WB (1:1000) |
| Antibody | Anti-Grb2 (rabbit monoclonal) | Santa Cruz | Cat#sc-255, RRID:AB_631602 | WB (1:1000) |
| Antibody | Anti-LRRC8A (rabbit polyclonal) | Pacific antibodies | Custom made | Epitope: QRTKSRIEQGIVDRSE, WB (1:1000), LRRC8A is a glucose sensor regulating ß-cell excitability and systemic glycemia (*Kang et al., 2018*) |
| Antibody | Anti-CD31 (rabbit polyclonal) | Thermo Fisher | Cat#MA3105, RRID:AB_223592 | IF (1:100) |
| Antibody | Anti-IgG (normal mouse IgG) | Santa Cruz | Cat#sc-2027, RRID:AB_737197 | WB (1:1000) |
| Antibody | Anti-rabbit-HRP | BioRad | Cat#170–6515, RRID:AB_11125142 | WB (1:10,000) |
| Antibody | Anti-mouse-HRP | BioRad | Cat#170–5047, RRID:AB_11125753 | WB (1:10,000) |
| Transfected construct (human) | si*Lrrc8a* | Invitrogen | Cat#4392420 | Assay ID: s32107 Transfected construct human |
| Transfected construct (human) | Non-targeting control | Invitrogen | Cat#4390846 | Transfected construct human |
| Transfected construct (human) | siPORTamine | Invitrogen | Cat#AM4503 | |
| Transfected construct (human) | Opti-MEM | Invitrogen | Cat#11058–021 | |
| Recombinant DNA reagent | Ad5-U6-h*Lrrc8a*-shRNA-mCherry | Vector biolabs | shADV-653 214592 | |
| Recombinant DNA reagent | Ad5-U6-*Scr*-shRNA-mCherry | Vector biolabs | 3086 | |
| Commercial assay or kit | RNA isolation (PureLink RNA mini kit) | Invitrogen | 12183018A | |
| Software, algorithm | GraphPad Prism8 | | La Jolla California USA, http://www.graphpad.com' RRID:SCR_002798 | |
| Software, algorithm | Fiji, ImageJ | *Schindelin et al., 2012* (PMID:22743772) | RRID:SCR_002285 | |
| Other | DAPI stain | Thermofisher | Cat#T3605 | 1:10,000 |

## Animals

The institutional animal care and use committee of the University of Iowa and Washington University School of Medicine approved all experimental procedures involving animals. All mice were housed in temperature, humidity, and light controlled environment and allowed water access and food. Both male and female *Lrrc8a*^fl/fl^ (*Zhang et al., 2017*; *Kang et al., 2018*) (WT), *CDH5-Cre;Lrrc8a*^fl/fl^ (*eLrrc8a*

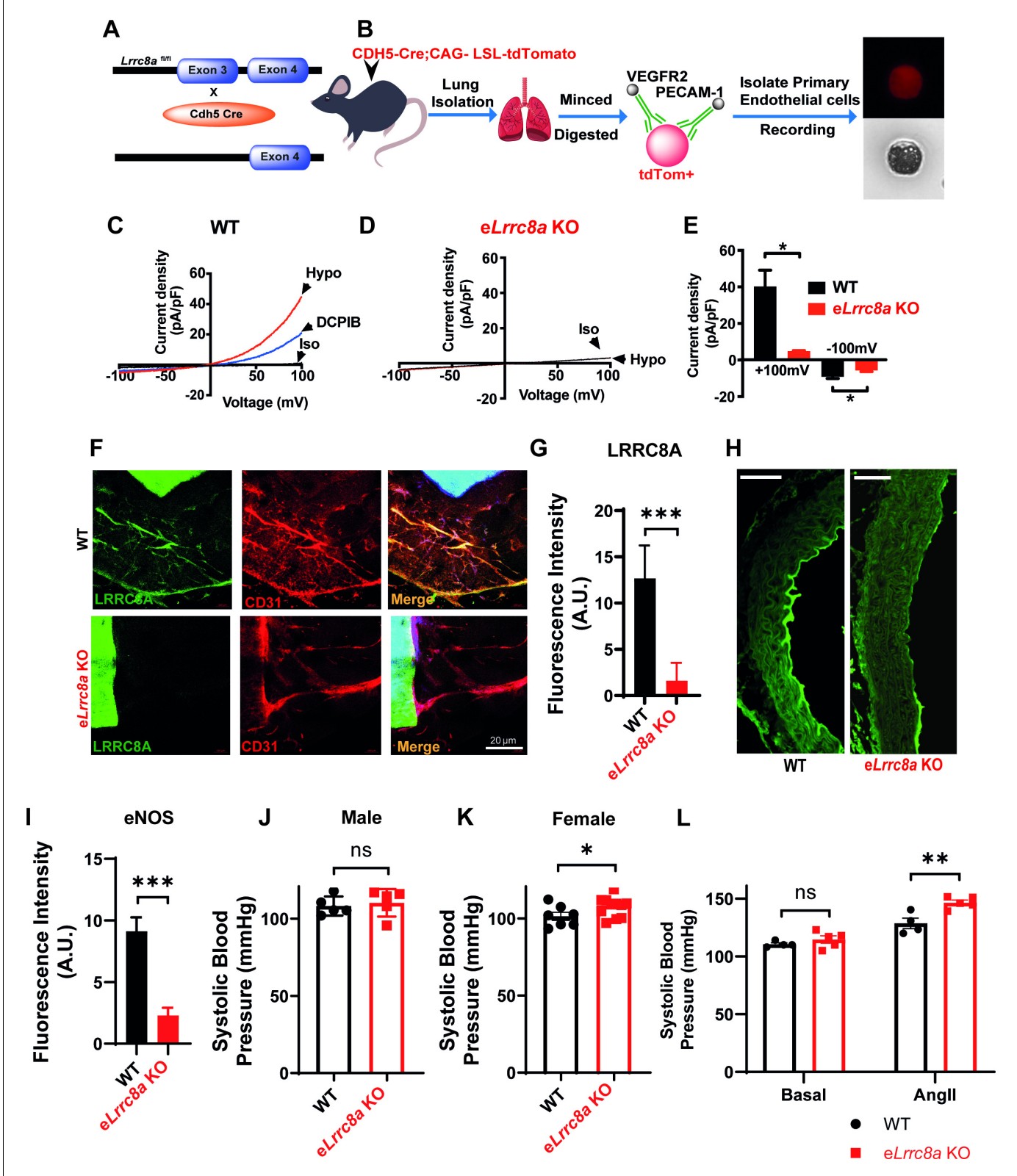

**Figure 7.** Endothelial-targeted *Lrrc8a* KO mice exhibit reduced phosphorylated endothelial nitric oxide synthase (p-eNOS) and mild angiotensin-II stimulated hypertension. (**A**) Strategy for endothelium targeted *Lrrc8a* ablation to generate e*Lrrc8a* KO. (**B**) Isolation of murine primary endothelial cells from WT and e*Lrrc8a* KO using tdTomato reporter mice. (**C–D**) Current-voltage relationships of volume-regulatory anion channel (VRAC) current in isotonic (Iso, 300 mOsM) and hypotonic (Hypo, 210 mOsm) solution in response to voltage ramps from -100 to +100 mV over 500 ms in WT (**C**) and KO
*Figure 7 continued on next page*

*Figure 7 continued*

(D) primary murine endothelial cells. DCPIB (10 μM) inhibition in C (WT). (E) Mean outward (+100 mV) and inward (-100 mV) currents from WT (n = 3 cells) and e*Lrrc8a* KO (n = 3 cells). (F) Ex vivo aorta sprouting assay performed in aortic rings isolated from WT and e*Lrrc8a* KO mice and cultured in FGM media for 3 days at 37°C. Immunofluorescence staining with antibodies to LRRC8A (green), CD31 (red), LRRC8A+CD31+ (Merge) demonstrating endothelial-targeted *Lrrc8a* deletion. (G) Quantification of LRRC8A immunofluorescence staining in WT (n = 15) and e*Lrrc8a* KO (n = 15) endothelial cell tubes. (H–I) Representative images of aortic sections from WT (n = 3) and e*Lrrc8a* KO (n = 3) mice immunostained with p-eNOS (H, green), and immunofluorescence quantification of regions of WT (n = 5) and e*Lrrc8a* KO (n = 5) endothelium (I). Scale bar in (H) is 50 μm. (J–L) Tail-cuff systolic blood pressures of male (J) and female (K) WT (n = 5 males and 7 females) and e*Lrrc8a* KO (n = 5 males and 12 females) mice, and systolic blood pressures of male WT (n = 4) and e*Lrrc8a* KO (n = 5) mice under basal conditions and after 4 weeks of chronic angiotensin-II infusion (L). Statistical significance between the indicated values is calculated using a two-tailed unpaired Student's t-test. Error bars represent mean ± s.e.m. *$p<0.05$; **$p<0.01$; ***$p<0.001$.

The online version of this article includes the following source data for figure 7:

**Source data 1.** Source data for *Figure 7E*.
**Source data 2.** Source data for *Figure 7I*.
**Source data 3.** Source data for *Figure 7J*.
**Source data 4.** Source data for *Figure 7K*.
**Source data 5.** Source data for *Figure 7I*.

KO) mice were generated and used in these studies. *CDH5-Cre* mice were obtained from Dr. Kaikobad Irani (University of Iowa, IA). In a subset of experiments, 5–8 week old *Lrrc8a*$^{fl/fl}$ and *CDH5-Cre; Lrrc8a*$^{fl/fl}$ mice were switched to HFHS (high-fat high-sucrose rodent diet, Research Diets, Inc, Cat# D12331) for at least 10 months. *CDH5-Cre* mice were crossed with *Rosa26-tdTomato* (Jax# 007914) reporter mice to identify CDH5+ cells for primary endothelium patch-clamp studies.

## Antibodies

Rabbit polyclonal anti-LRRC8A antibody was generated against the epitope QRTKSRIEQGIVDRSE (Pacific Antibodies) (*Kang et al., 2018*). All other primary antibodies were purchased from Cells Signaling: anti-ß-actin (#8457), Total Akt (#4685S), Akt1 (#2938), Akt2 (#3063), p-eNOS (#9571), Total eNOS (#32027), p-AS160 (#4288), p-p70 S6 Kinase (#9205S), pS6 Ribosomal (#5364S), GAPDH (#5174), pErk1/2 (#9101), Total Erk1/2 (#9102). Anti-LRRC8A antibody was custom made as described previously (*Zhang et al., 2017*; *Kang et al., 2018*). A second p-eNOS antibody was purchased from Invitrogen (Cat#PA5-104858) and used to compare with anti-p-eNOS from Cell Signaling (#9571) for p-eNOS IF staining. Purified mouse anti-Grb2 was purchased from BD (610111) and Santa Cruz (#sc-255). Rabbit IgG was purchased from Santa Cruz (sc-2027). Anti-CD31 was purchased from Thermo Fisher (MA3105).

## Electrophysiology

All recordings were performed in the whole-cell configuration at room temperature (RT), as previously described (*Zhang et al., 2017*; *Kang et al., 2018*). Briefly, currents were measured with either an Axopatch 200B amplifier or a MultiClamp 700B amplifier (Molecular Devices) paired to a Digidata 1550 digitizer. Both amplifiers used pClamp 10.4 software. The intracellular solution contained (in mM): 120 L-aspartic acid, 20 CsCl, 1 MgCl$_2$, 5 EGTA, 10 HEPES, 5 MgATP, 120 CsOH, 0.1 GTP, pH 7.2 with CsOH. The extracellular solution for hypotonic stimulation contained (in mM): 90 NaCl, 2 CsCl, 1 MgCl$_2$, 1 CaCl$_2$, 10 HEPES, 5 glucose, 5 mannitol, pH 7.4 with NaOH (210 mOsm/kg). The isotonic extracellular solution contained the same composition as above except for mannitol concentration of 105 (300 mOsm/kg). The osmolarity was checked by a vapor pressure osmometer 5500 (Wescor). Currents were filtered at 10 kHz and sampled at 100 μs interval. The patch pipettes were pulled from borosilicate glass capillary tubes (WPI) by a P-87 micropipette puller (Sutter Instruments). The pipette resistance was ~4–6 MO when the patch pipette was filled with intracellular solution. The holding potential was 0 mV. Voltage steps (500 ms) were elicited from 0 mV holding potential from -100 to +100 mV in 20 mV increments every 0.6 s. Voltage ramps from -100 to +100 mV (at 0.4 mV/ms) were applied every 4 s.

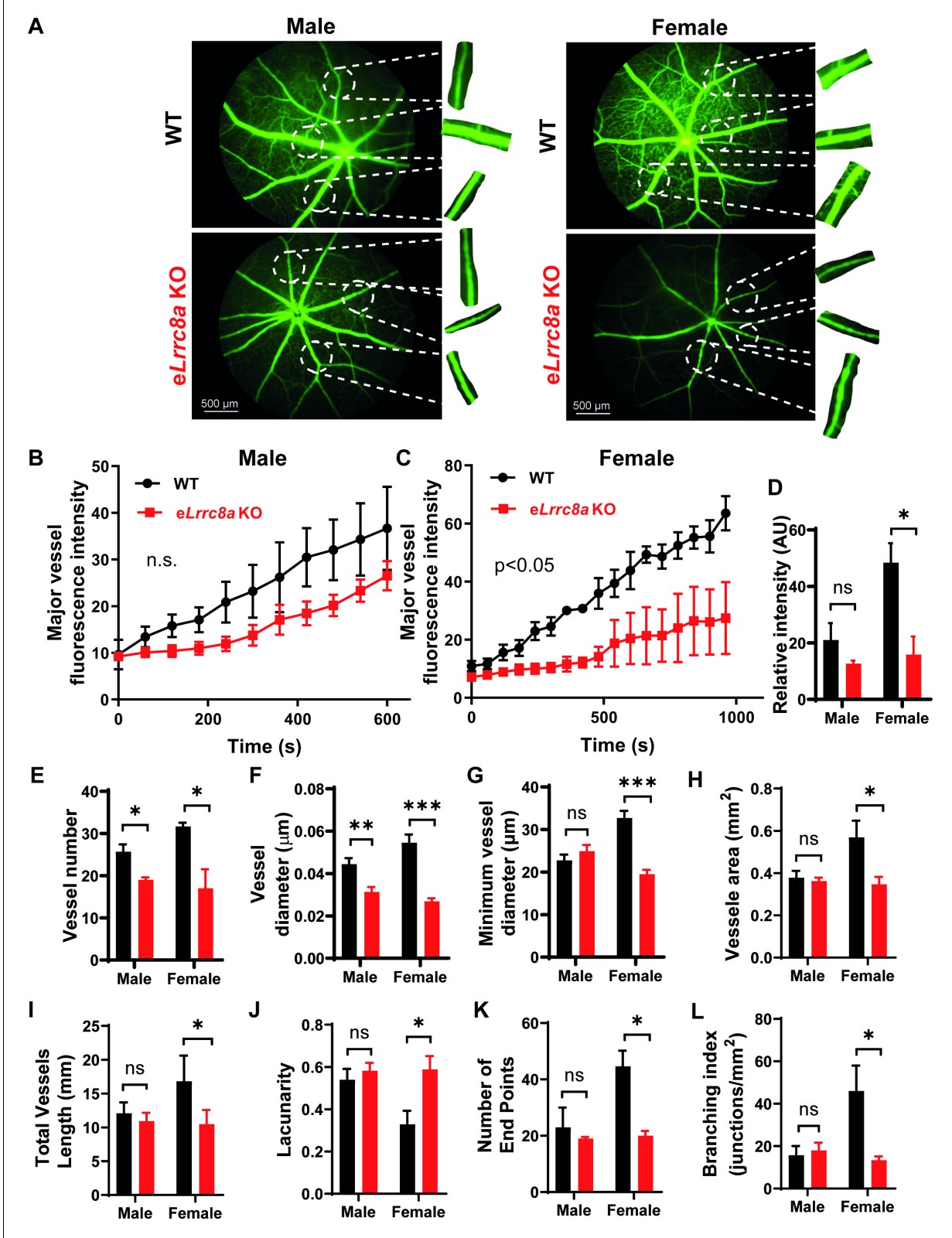

**Figure 8.** Endothelium-specific *Lrrc8a* KO mice exhibit exacerbated impairments retinal microvascular disease in the setting of type 2 diabetes. (**A**) Representative fluorescein retinal angiograms of WT (top) and e*Lrrc8a* KO (bottom) male (left) and female (right) mice raised on a high-fat high-sucrose (HFHS) diet. Inset shows magnified view of retinal vessels. Quantification of major vessel fluorescence intensity over time after intraperitoneal (i.p.) fluorescein injection in (**B**) male (n = 3) and (**C**) female (n = 3) WT and e*Lrrc8a* KO mice. (**D–K**) Quantification of total retinal vessel intensity (**D**), total

*Figure 8 continued on next page*

*Figure 8 continued*

vessel number (E), vessel diameter (F), minimum vessel diameter (G), vessel area (H), total vessel length (I), lacunarity (J), number of end points (K), and branching index (L) of retinal vessels in WT and e*Lrrc8a* KO mice. Statistical significance between the indicated values for **B** and **C** was calculated using two-way analysis of variance (ANOVA) and (**D–L**) using a two-tailed unpaired Student's t-test. Error bars represent mean ± s.e.m. *p<0.05; **p<0.01; ***p<0.001.

The online version of this article includes the following video, source data, and figure supplement(s) for figure 8:

**Source data 1.** Source data for *Figure 8B*.
**Source data 2.** Source data for *Figure 8C*.
**Source data 3.** Source data for *Figure 8D*.
**Source data 4.** Source data for *Figure 8E*.
**Source data 5.** Source data for *Figure 8F*.
**Source data 6.** Source data for *Figure 8G*.
**Source data 7.** Source data for *Figure 8H*.
**Source data 8.** Source data for *Figure 8I*.
**Source data 9.** Source data for *Figure 8J*.
**Source data 10.** Source data for *Figure 8K*.
**Source data 11.** Source data for *Figure 8L*.
**Figure supplement 1.** Endothelium-specific *Lrrc8a* KO mice exhibit mild retinal microvascular disease at baseline.
**Figure supplement 1—source data 1.** Source data for *Figure 8—figure supplement 1C*.
**Figure supplement 1—source data 2.** Source data for *Figure 8—figure supplement 1D*.
**Figure supplement 1—source data 3.** Source data for *Figure 8—figure supplement 1E*.
**Figure supplement 1—source data 4.** Source data for *Figure 8—figure supplement 1F*.
**Figure supplement 1—source data 5.** Source data for *Figure 8—figure supplement 1G*.
**Figure supplement 1—source data 6.** Source data for *Figure 8—figure supplement 1H*.
**Figure supplement 1—source data 7.** Source data for *Figure 8—figure supplement 1I*.
**Figure supplement 1—source data 8.** Source data for *Figure 8—figure supplement 1J*.
**Figure supplement 1—source data 9.** Source data for *Figure 8—figure supplement 1K*.
**Figure supplement 2.** Endothelium-specific *Lrrc8a* KO mice exhibit exacerbated impairments retinal microvascular disease in the setting of type 2 diabetes.
**Figure supplement 3.** Glucose tolerance (GTT) and insulin tolerance (ITT) are not altered in endothelium-specific *Lrrc8a* KO mice (n = 13) compared to WT mice (n = 8) raised on a high-fat high-sucrose diet for 10 months.
**Figure supplement 3—source data 1.** Source data for *Figure 8—figure supplement 3*.
**Figure supplement 4.** Body weight and body composition of WT and endothelium-specific *Lrrc8a* KO mice raised on a high-fat high-sucrose diet for 10 months.
**Figure supplement 4—source data 1.** Source data for *Figure 8—figure supplement 4*.
**Figure 8—video 1.** Retinal angiograms of WT (left) and e*Lrrc8a* KO (right) upon fluorescein injection intraperitoneally.
https://elifesciences.org/articles/61313#fig8video1

## Adenoviral KD

HUVECs were plated at 550,000 cell/well in 12-well plates. Cells were grown for 24 hr in the plates and transduced with either human adenovirus type 5 with sh*Lrrc8a* (sh*Lrrc8a*: Ad5-mCherry-U6-h*Lrrc8a*-shRNA, 2.2 × $10^{10}$ PFU/ml [shADV-214592], Vector Biolabs) or a scrambled non-targeting control (sh*Scr*: Ad5-U6-scramble-mCherry, 1 × $10^{10}$ PFU/ml) at a multiplicity of infection of 50 for 12 hr, and studies performed 3–4 days after adenoviral transduction. The shLRRC8a targeting sequence is: GCA CAA CAT CAA GTT CGA CGT.

## siRNA KD

HUVECs were plated at 360,000 cell/well in six-well plates. Cells were grown for 24 hr (90–95% confluency) and transduced with either a silencer select siRNA with si*LRRC8a* (Cat#4392420, sense: GCAACUUCUGGUUCAAAUUTT antisense: AAUUUGAACCAGAAGUUGCTG, Invitrogen) or a non-targeting control silencer select siRNA (Cat#4390846, Invitrogen), as described previously (*Koh et al., 2008*). The si*LRRC8a* used targets a different sequence from the shRNA described above. Briefly, siRNAs were transduced twice, 24 and 72 hr after HUVEC plating. Each siRNA was combined with Opti-MEM (285.25 µl, Cat#11058-021, Invitrogen) siPORT amine (8.75 µl, Cat#AM4503, Invitrogen) and the silencer select siRNA (6 µl) in a final volume of 300 µl. HUVECs were transduced over a 4 hr period at 37°C, using DMEM +1% FBS. After transduction, the cells

were returned to media containing M199, 20% FBS, 0.05 g heparin sodium salt, and 15 mg ECGS. Cell lysates were collected at basal conditions on day 4.

## Isolation of mouse lung endothelial cells

Isolation of mouse lung endothelial cells was performed according to the following protocol: Day 1 – Incubate sheep anti-rat IgG Dynabeads (Invitrogen) overnight with PECAM (Sigma, #SAB4502167) and VEGFR2 (R and D Systems, #BAF357) antibodies at 4°C in PBS with gentle agitation. Day 2 – Lungs were removed from the mice, washed in 10% FBS/DMEM, minced into 1–2 mm squares, and digested with Collagenase Type I (2 mg/ml, Gibco) at 37°C for 1 hr with agitation. The cellular digest was filtered through a 70 μm cell strainer, centrifuged at 1500 rpm, and the cells immediately incubated with the antibody coated Dynabeads at RT for 20 min. The bead-bound cells were recovered with a magnet, washed two times with PBS, and plated overnight on collagen type I (100 μg/ml) coated coverslips. The endothelial cells were maintained in a growth media of M199, 20% FBS, 0.05 g heparin sodium salt, 50 mg/ml ECGS, and 1× anti-anti.

## Cell culture

HUVECs were purchased from ATCC and were grown in MCDB-131-complete media overnight. HUVECs for basal condition collection were grown in growth media of M199, 20% FBS, 0.05 g heparin sodium salt (Cat#9041-08-1, Alfa Aesar), and 15 mg ECGS (Cat#02-102, Millipore Sigma). Cells were routinely cultured on 1% of gelatin-coated plates at 37°C at 5% $CO_2$. For insulin stimulation (Cat#SLBW8931), cells were serum starved for at least 13 hr in 1% FBS (Atlanta Bio selected, Cat#S11110) or without FBS using endothelial cells growth basal medium (Lonza Cat#cc-3121) instead of MCDB-131-complete media. Insulin stimulation was used for the times indicated at 100 nM.

## Immunoblotting

Cells were harvested in ice-cold lysis buffer (150 mM NaCl, 20 mM HEPES, 1% NP-40, 5 mM EDTA, pH 7.5) with added proteinase/phosphatase inhibitor (Roche). Cells were kept on ice with gentle agitation for 20 min to allow complete lysis. Lysate scraped into 1.5 ml tubes and cleared of debris by centrifugation at 14,000 × g for 20 min at 4°C. Supernatants were transferred to fresh tube and solubilized protein was measured using a DC protein assay kit (Bio-Rad). For immunoblotting, an appropriate volume of 1× Laemmli (Bio-Rad) sample loading buffer was added to the sample (10 μg of protein), which then heated at 90°C for 5 min before loading onto 4–20% gel (Bio-Rad). Proteins were separated using running buffer (Bio-Rad) for 2 hr at 150 V. Proteins were transferred to PVDF membrane (Bio-Rad) and membrane blocked in 5% (w/v) BSA in TBST or 5% (w/v) milk in TBST at RT for 2 hr. Blots were incubated with primary antibodies at 4°C overnight, followed by secondary antibody (Bio-Rad, Goat-anti-mouse #170–5047, Goat-anti-rabbit #170–6515, all used at 1:10,000) at RT for 1 hr. Membranes were washed three times and incubated in enhanced substrate Clarity (Bio-Rad) and imaged using a ChemiDoc XRS using Image Lab (Bio-Rad) for imaging and analyzing protein band intensities. ß-Actin or GAPDH levels were quantified to correct for protein loading.

## Immunoprecipitation

Cells were seeded on gelatin-coated 10 cm dishes in complete media for 24 hr. Adenoviruses, Ad5-mCherry-U6-h*Lrrc8a*-shRNA or Ad5-U6-*Scr*-mCherry were added to cells for 12 hr. After 4 days cells were serum starved for 16 hr with basal media containing 1% serum before stimulation with insulin (10 nM/ml). Cells were harvested in ice-cold lysis buffer (150 mM NaCl, 20 mM HEPES, 1% NP-40, 5 mM EDTA, pH 7.5) with added proteinase/phosphatase inhibitor (Roche) and kept on ice with gentle agitation for 15 min to allow complete lysis. Lysates were incubated with anti-Grb2 antibody (20 μg/ml) or control rabbit IgG (20 μg/ml, Santa Cruz sc-2027) rotating end over end overnight at 4°C. Protein G Sepharose beads (GE) were added to this for a further 4 hr before samples were centrifuged at 10,000 × g for 3 min and washed three times with RIPA buffer and re-suspended in laemmli buffer (Bio-Rad), boiled for 5 min, separated by SDS-PAGE gel followed by the Western blot protocol.

## Stretch assay

Equal amounts of cells were plated in each well in six well plated BioFlex coated with Laminin (BF-3001CCase) culture plate and seeded to ~90% confluence. Plates were placed into a FlexCell Jr. Tension System (FX-6000T) and incubated at 37°C with 5% $CO_2$. Prior to stretch stimulation, basal media of 1% FBS was added for 16 hr. Cells on flexible membrane were subjected to static stretch with the following parameter: a stretch of 0.1% and 5% with static strain. Cells were stretched for 5, 30, 60, or 180 min. Cells were then lysed and protein isolated for subsequent Western blots.

## Laminar flow studies

HUVECs were grown in 1× M199 media (Gibco, Cat#11043-023) supplemented with 20% FBS (Gibco, Cat#16140-071), 50 mg bovine hypothalamus extract, and 0.01% heparin until full confluence in six-well plates. The cells were transfected twice across 3 days with 100 nM of siRNA for the *Lrrc8a* gene (si*Lrrc8a*) (Invitrogen, Cat#4392421, assay #:s32107) and a negative siRNA control (siControl) (Invitrogen, Cat#43900846) using siPORTAmine transfection agent (Invitrogen, Cat#AM4503) following the manufacturer recommendations (*Koh et al., 2008*). Twenty-four hours after the final siRNA treatment, cells were transferred to μ-Slide 0.4 I Luer slide (iBidi, Cat#80176) and incubated overnight at 37°C, 5% $CO_2$, and 95% humidity. The slides were then cultured for 24 hr in a microscope stage incubator (Invitrogen, EVOS7000) under conditions of constant laminar flow at 15 dynes/cm²/s (Flocel Inc, Cat#QPL1010) and images acquired every 20 min. The peristaltic pump was converted to be laminar through the utilization of dampers on the outflow tracts of the pump. Efficacy in the generation of laminar flow was confirmed through the imaging of Q-Dots in the circulating media prior to the experiment (Invitrogen, Cat#Q21721MP). At the termination of the experiment, the cells were rinsed with 1× PBS and fixed with 4% paraformaldehyde (PFA) for downstream immunostaining applications.

## Immunostaining for laminar flow studies

To detect the levels of p-eNOS, the cell monolayer was stained with the anti-rabbit polyclonal antibody p-eNOS (Invitrogen, Cat#PA5104858 or Cell Signaling, Cat#9571), at a final dilution of 1:200 and a secondary anti-rabbit GFP AlexaFluor 488 (Invitrogen, Cat#R37116) at 1:2000 dilution. The slides were immunostained following the basic protocol: (1) 30 min RT incubation in Tris-Glycine; (2) 1 hr RT incubation ± permeabilization with 0.01% TritonX-100; (3) 2 hr RT incubation in blocking solution (5% Sheep Serum, 1% Roche Blocking Buffer in PBST); (4) 1 hr at RT – overnight 4°C incubation with 1:200 primary antibody unless otherwise noted; (5) wash with PBST; (6) 2–3 hr RT incubation with 1:2000 secondary antibody in 5% Sheep Serum, 1% Roche Blocking Buffer in PBST; (7) wash with PBST and imaging analysis. The samples were observed under light microscopy using a Nikon Ti2. Quantification of immunostaining intensity was performed using ImageJ/Fiji (*Schindelin et al., 2012*).

## Immunofluorescence imaging

Cells were plated on gelatin-coated glass coverslips. Cells on coverslip were washed in PBS and fixed with 2% (w/v) PFA for 20 min at RT. PFA was washed three times with PBS and permeabilized in PBS containing 0.2% Triton X-100 for 5 min at RT. Cells on coverslips were washed in PBS and blocked for 30 min at RT with TBS containing 0.1% Tween-20% and 5% BSA. Cells on coverslip were incubated overnight at 4°C with primary antibody (1:250) in TBS containing 0.1% Tween-20% and 1% BSA. Cells were then washed in PBS five times and incubated for 2 hr at RT with 5% BSA in TBST. Cells were washed three times in TBST and then incubated with secondary antibodies at 1:1000 dilution (Invitrogen, Anti-Rabbit 488, A11070; Anti-mouse 568, A11019; Anti-mouse 488, A11017) for 1 hr at RT. Coverslips were then incubated for 10 min with Topro 3 (T3605, Thermofisher) or mounted with mounting media containing DAPI (Invitrogen) to visualize nuclei. Images were taken using Axio-cam 503 Mono Camera controlled by Zeiss Blue using a Plan-Apochromate 40× oil immersion objective.

## Immunohistochemistry of thoracic aorta sections

Mouse thoracic aortas were dissected and immediately fixed in 10% formalin. The formalin fixed aortas were embedded in paraffin and 5 μm sections were cut by automated microtome and mounted

on positively charged slides. Slides were deparaffinized with Histoclear and hydrated with 100% ethanol, 75% ethanol, 50% ethanol, and then distilled water. To achieve the antigen retrieval, slides were steamed in Citrate buffer at pH 6 (DAKO Target retrieval buffer S1699, DAKO) for 30 min and subsequently cooled for 30 min. Furthermore, to inhibit the endogenous peroxides, slides were quenched with 3% $H_2O_2$ for 10 min. Slides were washed with distilled water, then $1\times$ PBS and blocked with 1% BSA in $1\times$ PBS for 1 hr. After washing with $1\times$ PBS, slides were incubated with p-eNOS antibody (1:500 dilution) overnight in the humidified chamber at 4°C. Slides were washed two times with $1\times$ PBS and incubated with secondary antibody (AF488, Invitrogen, 1:1000 dilution) for 1 hr. Subsequently slides were washed with $1\times$ PBS three times and mounted with DAPI added mounting media with coverslip. The slides were imaged by Zeiss LSM700 confocal microscope ($10\times$ objective).

## Ex vivo sprouting angiogenesis

Following Avertin injection and cervical dislocation, aortas were dissected and connective tissue removed, and then washed with PBS with 50 µg/ml penicillin and streptomycin. Using iris scissors, the aorta was cut into aortic rings of 1– 2 mm cross-sectional slices; 50 µl of Matrigel was used to coat the center of coverslips in 24-well plates for 2 hr at 37°C in the incubator to solidify the Matrigel. Aorta rings were then seeded and transplanted on Matrigel (BD Biosciences, Cat#356231) on coverslips. After seeding the aortic rings, plates were incubated at 37°C without medium for 10 min to allow the ring to attach to the Matrigel. Complete medium was added to each well and incubated at 37°C with 5% $CO_2$ for 48–72 hr. Phase contrast photos of individual explants were taken using a $10\times/0.75$ NA objective Olympus IX73 microscope (Olympus, Japan) fitted with camera (Orca flash 4.0+, Hamamatsu, Japan). Cells were incubated with LRRC8A (1:250) and CD31 (1:250) primary antibodies in 0.1% Tween-20% and 1% BSA overnight, and then incubated with secondary antibodies (Invitrogen, Anti-Rabbit 488, A11070; Anti-Mouse 568, A11019). Cells were then incubated for 10 min with Topro 3 (T3605, Thermofisher) at RT.

## Retinal imaging

Ketamine (Akorn Animal Health, 100 mg/ml) was prepared and mixed with xylazine, then stored at RT. Animals were anesthetized with 87.5/12 mg/kg body weight via i.p. injection. Eyes were topically anesthetized with proparacaine and dilated with tropicamide. Fluorescein (100 mg/ml, Akorn Inc) diluted with sterile saline was administered by i.p. injection (50 µl), and mice were positioned on Micron imaging platform. Images of the eyes were taken from the start of fluorescein infusion with the Micron camera with a 450–650 nm excitation filter and 469–488 nm barrier filter at 30 frame/s using Micron software for 30 s. Data were converted into tiff image for further analysis.

## Ang II infusion

Infusion studies were carried out using Azlet osmotic minipumps (Model 1004). Ang II (BACHEM) dissolved in saline was filled in the minipumps and were prepared to maintain infusion rate of 600 ng/kg/min for 4 weeks. The mice were anesthetized under 2% isofluorane and the minipumps were implanted subcutaneously on the dorsal aspect of the mice.

## Blood pressure recordings

Systolic tail-cuff blood pressure (BP) measurements were carried out using computerized tail-cuff system BP-2000 (Visitech Systems) at the same time of day. Mice were first acclimated to the device by performing 3 days of measurements (20 sequential measurements/day) and then mean blood pressure readings were obtained by averaging 3–5 days of measurements (not inclusive of the 3 acclimation days).

## RNA sequencing

RNA quality was assessed by Agilent BioAnalyzer 2100 by the University of Iowa Institute of Human Genetics, Genomics Division. RNA integrity numbers greater than eight were accepted for RNAseq library preparation. RNA libraries of 150 bp PolyA-enriched RNA were generated, and sequencing was performed on a HiSeq 4000 genome sequencing platform (Illumina). Sequencing results were uploaded and analyzed with BaseSpace (Illumina). Sequences were trimmed to 125 bp using FASTQ

Toolkit (Version 2.2.0) and aligned to *Homo sapiens* HG19 (refseq) using RNA-Seq Alignment (Version 1.1.0). Transcripts were assembled and differential gene expression was determined using Cufflinks Assembly and DE (Version 2.1.0). Ingenuity Pathway Analysis (QIAGEN) was used to analyze significantly regulated genes which were filtered using cutoffs of >1.5 fragments per kilobase per million reads, >1.5-fold changes in gene expression, and a false discovery rate of <0.05. Heatmaps were generated to visualize significantly regulated genes. Data have been deposited in GEO (accession# TBD).

## Metabolic phenotyping

Mice were fasted for 6 hr prior to glucose tolerance tests (GTTs). Baseline glucose levels at 0 min timepoint (fasting glucose) were measured from blood samples collected from tail bleeds using a glucometer (Bayer Healthcare LLC). D-Glucose was injected i.p. (0.75 g/kg body weight) and glucose levels were measured at 7, 15, 30, 60, 90, and 120 min timepoints after injection. For insulin tolerance tests (ITTs), the mice were fasted for 4 hr. Similar to GTTs, the baseline blood glucose levels were measured at 0 min timepoint and 15, 30, 60, 90, and 120 min timepoints post-injection (i.p.) of insulin (HumulinR, 1.25 U/kg body weight). Mouse body composition was performed as previously described (*Zhang et al., 2017*).

## Statistics

Data are represented as mean ± s.e.m. Two-tail unpaired Student's t-tests were used for comparison between two groups. For three or more groups, data were analyzed by one-way analysis of variance (ANOVA) and Tukey's post hoc test. For time-series data in *Figure 5* between two groups, data were analyzed using a two-way ANOVA followed by Bonferroni post hoc test. For GTTs and ITTs, two-way ANOVA was used. A p-value<0.05 was considered statistically significant. *, **, and *** represent a p-value less than 0.05, 0.01, and 0.001, respectively.

# Acknowledgements

RNA-Seq data presented herein were obtained at the Genomics Division of the Iowa Institute of Human Genetics. This work was supported by grants from the NIH/NHLBI R01 HL125436 (CEG), NIH/NHLBI 5R00HL125683, NIH/NIGMS R35 GM137976, Cancer Research Foundation Young Investigator Award (ANS), NIH NIDDK 1R01DK106009, the Roy J Carver Trust (RS), UIHC Center for Hypertension Research Pilot and Feasibility Grant, VA Merit Award I01BX005072 (RS), andKing Abdullah International Medical Research Center (KAIMRC) grant RA17-014-A (AA). We thank Dr. Rithwick Rajagopal for insightful reading of the manuscript.

# Additional information

## Funding

| Funder | Grant reference number | Author |
|---|---|---|
| National Institute of Diabetes and Digestive and Kidney Diseases | R01DK106009 | Rajan Sah |
| National Heart, Lung, and Blood Institute | 5R00HL125683 | Amber N Stratman |
| National Heart, Lung, and Blood Institute | R01 HL125436 | Chad E Grueter |
| National Heart, Lung, and Blood Institute | R35 GM137976 | Amber N Stratman |
| VA Merit Award | I01BX005072 | Rajan Sah |
| Cancer Research Foundation | Young Investigator Award | Amber N Stratman |
| Roy J. Carver Charitable Trust | | Rajan Sah |
| UIHC Center | I01B 005072 | Rajan Sah |

The funders had no role in study design, data collection and interpretation, or the decision to submit the work for publication.

## Author contributions
Ahmad F Alghanem, Data curation, Software, Formal analysis, Supervision, Validation, Investigation, Visualization, Methodology, Writing - review and editing; Javier Abello, Investigation, Visualization, Methodology; Joshua M Maurer, Chau My Ta, Formal analysis, Investigation, Visualization, Methodology, Writing - review and editing; Ashutosh Kumar, Formal analysis, Supervision, Investigation, Methodology; Susheel K Gunasekar, Formal analysis, Supervision, Investigation, Visualization, Methodology; Urooj Fatima, Oluwaseun Adeola, Formal analysis, Investigation, Methodology; Chen Kang, Formal analysis, Visualization, Methodology; Litao Xie, Robert F Mullins, Amber N Stratman, Supervision, Methodology; Megan Riker, Rachel A Minerath, Methodology; Macaulay Elliot-Hudson, Formal analysis, Methodology; Chad E Grueter, Data curation, Formal analysis, Supervision, Methodology; Rajan Sah, Conceptualization, Data curation, Formal analysis, Supervision, Funding acquisition, Validation, Methodology, Writing - original draft, Project administration, Writing - review and editing

## Author ORCIDs
Ahmad F Alghanem (iD) https://orcid.org/0000-0002-5823-3806
Amber N Stratman (iD) https://orcid.org/0000-0002-8111-4186
Rajan Sah (iD) https://orcid.org/0000-0003-1092-1244

## Ethics
Animal experimentation: This study was performed in accordance with the recommendations in the Guide for the Care and Use of Laboratory Animals of the National Institutes of Health. All of the animals were handled according to the approved Institutional Animal Care and Use Committee (IACUC) protocols of Washington University in St. Louis (20180217) and the University of Iowa (1308148).

## Decision letter and Author response
Decision letter https://doi.org/10.7554/eLife.61313.sa1
Author response https://doi.org/10.7554/eLife.61313.sa2

# Additional files
## Supplementary files
• Supplementary file 1. RNA sequencing data of human umbilical vein endothelial cell (HUVEC) treated with Ad-sh*Scr* and Ad-sh*Lrrc8a.*

• Supplementary file 2. Ingenuity pathway analysis.

• Transparent reporting form

## Data availability
RNA sequencing data has been deposited in GEO under accession number GSE166989. All other data available as source data files.

The following dataset was generated:

| Author(s) | Year | Dataset title | Dataset URL | Database and Identifier |
|---|---|---|---|---|
| Alghanem Af, Abello J, Maurer JM, Kumar A, Ta C, Gunasekar SK, Fatima U, Kang C, Xie L, Adeola O, Riker M, Elliott-Hudson M, | 2021 | The SWELL1-LRRC8 complex regulates endothelial AKT- eNOS- mTOR signaling and vascular function | https://www.ncbi.nlm. nih.gov/geo/query/acc. cgi?acc=GSE166989 | NCBI Gene Expression Omnibus, GSE166989 |

Minerath RA,
Grueter CE, Mullins
RF, Stratman AN,
Sah R

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
