## [Decision Letter]

**Acceptance summary:**

In their manuscript, Alghanem et al. provide the first demonstration that LRRC8a (leucine rich repeat-containing protein 8a), also known as SWELL1, encodes the endothelial volume regulatory anion channel (VRAC). Using a human umbilical vein endothelial cell (HUVEC) model and an RNA interference (RNAi) approach, they also show that SWELL1 knockdown in vitro suppresses AKT-eNOS signaling while enhancing mTOR signaling, suggesting that SWELL1 normally positively regulates the AKT-eNOS pathway and negatively regulates mTOR signaling.

**Decision letter after peer review:**

Thank you for submitting your article "The SWELL1-LRRC8 complex regulates endothelial AKT-eNOS-mTOR signaling and vascular function" for consideration by *eLife*. Your article has been reviewed by three peer reviewers, including Mark T Nelson as the (Reviewing Editor and Reviewer #1), and the evaluation has been overseen by Kenton Swartz as the Senior Editor. The following individual involved in review of your submission has agreed to reveal their identity: Piruthivi Sukumar (Reviewer #3).

The reviewers have discussed the reviews with one another and the Reviewing Editor has drafted this decision to help you prepare a revised submission.

Summary:

In their manuscript, Alghanem et al. provide the first demonstration that LRRC8a (leucine rich repeat-containing protein 8a), also known as SWELL1, encodes the endothelial volume regulatory anion channel (VRAC). Using a human umbilical vein endothelial cell (HUVEC) model and an RNA interference (RNAi) approach, they also show that SWELL1 knockdown in vitro suppresses AKT-eNOS signaling while enhancing mTOR signaling, suggesting that SWELL1 normally positively regulates the AKT-eNOS pathway and negatively regulates mTOR signaling. Consistent with previous results obtained in adipocytes, they also found that endothelial SWELL1 forms a signaling complex with GRB2, Cav1 and eNOS, and further showed that this SWELL1-GRB2-Cav1-eNOS complex mediates stretch-induced activation of PI3K-AKT-eNOS signaling. in vivo studies in endothelial-specific Swell1-knockdown mice confirmed that SWELL1 mediates hypotonically activated currents in native endothelial cells and also negatively regulates endothelial cell migration and angiogenesis, as evidenced by an increase in ex vivo sprouting angiogenesis from aortic explants from mice lacking endothelial SWELL1. Consistent with this, genome-wide transcriptome analyses showed enrichment for genes/pathways involved in regulating angiogenesis, migration and tumorigenesis, including GADD45, IL-8, p70S6K (mTOR), TREM1, angiopoeitin and HGF, in SWELL1-KD HUVECs. Knockdown of SWELL1 in HUVECs also increased expression of VEGFA and CD31, both of which are pro-angiogenic and are associated with mTORC1 hyperactivation. Finally, given previous reports of SWELL1 involvement in insulin signaling (via PI3K-AKT) in adipocytes and overrepresentation of pathways (e.g., cell adhesion, renin-angiotensin signaling) in SWELL1-KD HUVECs that are altered in the vasculature in the context of atherosclerosis and type 2 diabetes (T2D), they investigated the effects of SWELL1 knockdown in a hypertensive (angiotensin II infusion) and T2D (high-fat high-sucrose diet) setting, demonstrating exacerbated systolic hypertension and impaired retinal blood flow in male SWELL1-KD mice.

1) The authors should respond to concerns with careful editing of the manuscript and refrain from conclusions not fully supported by the current data, such as "SWELL1 regulates AKT-eNOS signaling via SWELL1-GRB2-Cav1 signaling complex", "SWELL1 negatively regulates mTOR signaling". In addition, just like hypotonicity-activated TRPV4, whether VRAC is mechanosensitive is also subject to debate.

2) The evidence for the interaction of SWELL1 with Grb2, Cav1 and eNOS does not necessarily reveal the underlying mechanism(s) for how SWELL1 regulates the basal eNOS expression and AKT-eNOS signaling. What might activate/regulate SWELL1 at the basal condition? Do the effects require the SWELL1 channel activity? If yes, how the channel conduction regulates Grb2 signaling? If not, how the protein-protein interaction itself regulates Grb2 signaling. Knocking down SWELL1 did not reduce the interaction between Grb2 and Cav1 (Figure 3A). So presumably, SWELL1 is not required for the protein complex formation per se. The data presented does not address the mechanism on how SWELL1 regulates eNOS expression and the basal AKT-eNOS signaling.

3) Fluid shear stress is a more relevant force experienced by endothelium. It's well-established that shear stress stimulates many folds increase in phosphorylation of AKT and eNOS in HUVEC. It's one of the hallmarks of endothelial mechanosensitivity. However, the authors performed experiments by stretching HUVEC cells and accordingly observed very poor AKT-eNOS response. Thus, shear stress might be a more appropriate system to test the potential role of SWELL1 in endothelium mechanotransduction. It's still very unclear whether SWELL1 is indeed a mechanosensitive channel. Is endothelial SWELL1 channel activated by shear stress and/or stretch? How does the channel activation lead to downstream changes in AKT-eNOS signaling, by changes in membrane potential, secondary massager, or changes in Grb2-Cav1-eNOS complexes?

4) The authors suggested that SWELL1 regulates AKT-eNOS, NO production, and thus play a role in vasodilation, which may explain the KO mouse phenotype, such as hypertension and retinal microvascular disease in the setting of Type 2 diabetes. A more detailed biochemical and cell biological characterization of eSWELL1 KO mice is warranted to bridge the data in HUVEC with the mouse model. For example, it's well-establish that flow induces vasodilation of mesenteric arteries ex vivo. Do eSWELL1 KO arteries exhibit a flow-induced vasodilation defect? Does the dramatic downregulation of eNOS expression exist in KO mice? Does the flow-induced NO release is impaired in KO endothelium?

5) Total eNOS and AKT level changes are interesting and which are neither explored experimentally nor discussed. Authors have to comment on the potential mechanism/significance of that with respect to phenotypic observations.

6) Is SWELL1 constitutively active? How do the authors explain the "basal" (without stretch) effects of SWELL1 KD?

7) Both chow/HFHS model animals with SWELL1 KO were not phenotyped completely. At least basal characteristics such as weight, activity etc should be reported to rule out the "metabolic" causes for observed vascular effects. It is especially important since the ITT/GTT were done after 10 months- an advanced/old stage in which it is hard to see differences in ITT/GTT.

8) mTOR pathway is too underdeveloped to make any claim in the Abstract; I think it is mere observation which could be due to "unknown" signalling outside of what is described in the manuscript.

9) Explant assay data presented in Figure 5 is distracting as it does not fit with eNOS story; Expectedly the SWELL1 signalling is more complex than reported- both pro and anti angiogenic. But unless it is explored bit deeper, it won't add anything to the manuscript.

10) Finally, there is another mechanically activated channel in EC- Piezo1. What was its' level in KO cells (RNA seq data)? Some discussion on that will be useful for the readers.

---

## [Author Response]

Revisions for this paper:1) The authors should respond to concerns with careful editing of the manuscript and refrain from conclusions not fully supported by the current data, such as "SWELL1 regulates AKT-eNOS signaling via SWELL1-GRB2-Cav1 signaling complex", "SWELL1 negatively regulates mTOR signaling". In addition, just like hypotonicity-activated TRPV4, whether VRAC is mechanosensitive is also subject to debate.

We agree with these comments and thank the reviewers for pointing this out. We have edited the manuscript as described above.

2) The evidence for the interaction of SWELL1 with Grb2, Cav1 and eNOS does not necessarily reveal the underlying mechanism(s) for how SWELL1 regulates the basal eNOS expression and AKT-eNOS signaling. What might activate/regulate SWELL1 at the basal condition? Do the effects require the SWELL1 channel activity? If yes, how the channel conduction regulates Grb2 signaling? If not, how the protein-protein interaction itself regulates Grb2 signaling. Knocking down SWELL1 did not reduce the interaction between Grb2 and Cav1 (Figure 3A). So presumably, SWELL1 is not required for the protein complex formation per se. The data presented does not address the mechanism on how SWELL1 regulates eNOS expression and the basal AKT-eNOS signaling.

We agree that we have not delineated the detailed molecular mechanism that underlies how SWELL1 regulates basal eNOS expression, nor how it regulates eNOS phosphorylation (p-eNOS). In this paper, we merely demonstrate that SWELL1 forms a complex with GRB2, Cav1 and eNOS, demonstrate that SWELL1 depletion impairs pAKT and peNOS, and then hypothesize (now added in the Discussion) that these signaling effects occur via protein-protein interactions between SWELL1, GRB2, Cav1 and eNOS – similar to our previously proposed working model for SWELL1-GRB2-AKT2 signaling in adipocytes (Zhang et al., 2017 and Gunasekar et al., 2019). We also propose an alternative mechanism that considers SWELL1 channel mediated effects in endothelial cell membrane and calcium influx via other mechanosensitive (Piezo1) or mechanoresponsive (TRPV4) – also added to the Discussion section. Delineating these detailed molecular mechanisms will be the subject of future manuscripts.

3) Fluid shear stress is a more relevant force experienced by endothelium. It's well-established that shear stress stimulates many folds increase in phosphorylation of AKT and eNOS in HUVEC. It's one of the hallmarks of endothelial mechanosensitivity. However, the authors performed experiments by stretching HUVEC cells and accordingly observed very poor AKT-eNOS response. Thus, shear stress might be a more appropriate system to test the potential role of SWELL1 in endothelium mechanotransduction.

The vascular endothelium is exposed to a variety of mechanical forces, including laminar, continuous, turbulent and pulsatile shear flow, in addition to stretching forces that occur during arterial pressure waves with each heartbeat. Indeed, there are dozens of papers that have studied the response of endothelial cells to axial stretch (Reviewed in Jufri et al., 2015) as we have, so it remains, in our opinion, a relevant mechanical stimulus to study. However, we agree that shear flow is also a relevant stimulus to examine in HUVECs upon SWELL1 depletion. Accordingly, we have included a new experiment, shown in new Figure 5 and new Figure 5—videos 1-4, wherein we discover that the ability of HUVECs to align along the direction of laminar shear flow is markedly impaired in SWELL1 KD cells – suggesting that these cells have either impaired ability to sense, or to respond to laminar shear flow (or both). Associated with this is a reduction in p-eNOS induction (assessed by IF staining) in response shear flow in SWELL1 depleted HUVECs. These data are consistent with the p-eNOS results observed under basal conditions, and in response to axial stretch. The shear flow system available to us for these studies was limited in the number of cells that could be plated, and thus we were unable to perform Western blots to assess for shear-flow mediated signaling as we could perform with the FlexCell system, despite weeks of effort. It is for this reason that we resorted to quantification of peNOS by immunofluorescence staining – which is feasible with a much smaller number of cells. Regardless, we thank the reviewer for raising this question as it allowed us to uncover a cellular phenotype that we would not have otherwise observed, and also motivated us to explore another physiologically relevant form of mechanical stimulation.

It's still very unclear whether SWELL1 is indeed a mechanosensitive channel. Is endothelial SWELL1 channel activated by shear stress and/or stretch? How does the channel activation lead to downstream changes in AKT-eNOS signaling, by changes in membrane potential, secondary massager, or changes in Grb2-Cav1-eNOS complexes?

These are both very good questions. We have not answered these specific questions in this manuscript, but they remain questions for the next paper, or for a Research Advance, as we are currently developing methods to measure endothelial SWELL1 channel activation in response to shear flow. The hypotheses addressing the second question of how “channel activation leads to downstream changes in AKT-eNOS signaling, by changes in membrane potential, secondary massager, or changes in Grb2-Cav1-eNOS complexes” are also now addressed in the Discussion.

4) The authors suggested that SWELL1 regulates AKT-eNOS, NO production, and thus play a role in vasodilation, which may explain the KO mouse phenotype, such as hypertension and retinal microvascular disease in the setting of Type 2 diabetes. A more detailed biochemical and cell biological characterization of eSWELL1 KO mice is warranted to bridge the data in HUVEC with the mouse model. For example, it's well-establish that flow induces vasodilation of mesenteric arteries ex vivo. Do eSWELL1 KO arteries exhibit a flow-induced vasodilation defect? Does the dramatic downregulation of eNOS expression exist in KO mice? Does the flow-induced NO release is impaired in KO endothelium?

As mentioned in the introductory paragraph above, new experiments in the eSWELL1 KO mouse were not possible during these past few months due to a temporary lack of availability of this mouse model, arising from the impact of COVID19. Accordingly, we have addressed these comments “to bridge the data in HUVEC with the mouse model” by using tissue samples that were previously collected from these mice. Specifically, we performed immunofluorescence staining for p-eNOS in endothelium from aortas harvested from WT and eSWELL1 KO mice (Figure 7H and I), and found this to be diminished as well, similar to SWELL1 KD HUVECs. Again, once a colony of WT and eSWELL1 KO mice again become available, we will examine flow-mediated vasodilation and other measures of flow-induced NO release in vascular endothelium.

5) Total eNOS and AKT level changes are interesting and which are neither explored experimentally nor discussed. Authors have to comment on the potential mechanism/significance of that with respect to phenotypic observations.

We agree that these changes in eNOS expression are interesting and warrant further exploration. We can only speculate that SWELL1 is somehow regulating eNOS gene expression. Given that NF-kappaB is thought to regulate eNOS expression in endothelium we hypothesize that this pathway may be implicated. Moreover, both PI3K and ERK1/2 signaling pathways have been described to regulate total eNOS expression (Reviewed in Wu et al., 2002), and we observe reductions in both PI3K and ERK1/2 signaling in SWELL1 KD HUVECs, so reductions in eNOS expression are certainly consistent with disruptions in these signaling pathways. Indeed, consistent with this conserved biology with respect to SWELL1 signaling, we also observed significant 2-fold reductions in AKT2 expression upon SWELL1 deletion in both 3T3-F442A adipocytes and in C2C12 myotubes, in addition to reductions in NOS2 (iNOS) in adipocytes and NOS1 (neuronal NOS) in C2C12. Collectively, these findings across multiple cell types support a PI3K and ERK1/2 mediated mechanism of regulation of eNOS and AKT2 gene expression. These mechanisms are now discussed in the Discussion.

6) Is SWELL1 constitutively active? How do the authors explain the "basal" (without stretch) effects of SWELL1 KD?

We do believe that SWELL1 has constitutive activity, via protein-protein signaling, by restraining GRB2 inhibition of the PI3K-AKT-MAPK pathway, or by conductive, channel mediated mechanisms that are yet to be explored. Indeed, others have made measurement of basal, or constitutive VRAC activity in neurons (Zhang, H., et al., PLoS ONE 2011; Inoue, H., et al., J. Neurosci 2007) and ß cells (Best, L. et al. Pflug. Arch. 2002) and estimate it at NP = ~0.06, at 1 mM glucose in ß cells.

7) Both chow/HFHS model animals with SWELL1 KO were not phenotyped completely. At least basal characteristics such as weight, activity etc should be reported to rule out the "metabolic" causes for observed vascular effects. It is especially important since the ITT/GTT were done after 10 months- an advanced/old stage in which it is hard to see differences in ITT/GTT.

We now include additional metabolic phenotyping data of the WT and eSWELL1 KO mice raised on HFHS diet, including body weight, fat mass, lean mass, % fat mass and % lean mass in Figure 8—figure supplement 4 to complement the GTT and ITT data provided in the first submission. These data reveal a difference in body weight that is driven by increased adiposity in female but not male mice. However, the significant differences in retinal vessel blood flow in female mice are observed in the absence of measurable differences in glycemic control or systemic insulin sensitivity.

8) mTOR pathway is too underdeveloped to make any claim in the Abstract; I think it is mere observation which could be due to "unknown" signalling outside of what is described in the manuscript.

We agree completely with this comment and have accordingly removed this from the Abstract and primarily describe the mTOR pathway findings as a curious observation.

9) Explant assay data presented in Figure 5 is distracting as it does not fit with eNOS story; Expectedly the SWELL1 signalling is more complex than reported- both pro and anti angiogenic. But unless it is explored bit deeper, it won't add anything to the manuscript.

We agree and appreciate this feedback in helping us streamline the message of this manuscript. Accordingly, we have removed the explant data analysis from the manuscript, but have left in quantification of SWELL1 immunostaining of WT compared to eSWELL1 KO endothelial tubes, since these data provide validation of the endothelial targeted SWELL1 KO mouse (in addition to the patch-clamp data from WT and SWELL1 KO primary endothelial cells shown in Figure 7A-E).

10) Finally, there is another mechanically activated channel in EC- Piezo1. What was its' level in KO cells (RNA seq data)? Some discussion on that will be useful for the readers.

Indeed, EC-Piezo1 is a very important mechanically activated ion channel to discuss. In response to this comment, we now formally plot the Piezo1 mRNA levels in HUVEC (+/- SWELL1) obtained from our RNA sequencing experiments (as well at TRPV4, another very relevant mechanoresponsive ion channel expressed in endothelium), and compare this to SWELL1 mRNA expression and other LRRC8 proteins in new Figure 6C. We also add some discussion in the Results and Discussion.